# HA-ViD: A Human Assembly Video Dataset for Comprehensive Assembly Knowledge Understanding

**Hao Zheng**[*]
hzhe951@aucklanduni.ac.nz

**Regina Lee**[*]
klee702@aucklanduni.ac.nz

**Yuqian Lu**[†]
yuqian.lu@auckland.ac.nz
Department of Mechanical and Mechatronics Engineering
The University of Auckland

## Abstract

Understanding comprehensive assembly knowledge from videos is critical for futuristic ultra-intelligent industry. To enable technological breakthrough, we present HA-ViD – an assembly video dataset that features representative industrial assembly scenarios, natural procedural knowledge acquisition process, and consistent human-robot shared annotations. Specifically, HA-ViD captures diverse collaboration patterns of real-world assembly, natural human behaviors and learning progression during assembly, and granulate action annotations to subject, action verb, manipulated object, target object, and tool. We provide 3222 multi-view and multi-modality videos, 1.5M frames, 96K temporal labels and 2M spatial labels. We benchmark four foundational video understanding tasks: action recognition, action segmentation, object detection and multi-object tracking. Importantly, we analyze their performance and the further reasoning steps for comprehending knowledge in assembly progress, process efficiency, task collaboration, skill parameters and human intention. Details of HA-ViD is available at: https://iai-hrc.github.io/ha-vid.

## 1 Introduction

Assembly knowledge understanding from videos is crucial for futuristic ultra-intelligent industrial applications, such as robot skill learning [1], human-robot collaborative assembly [2] and quality assurance [3]. To enable assembly video understanding, a video dataset is required. Such a video dataset should (1) represent real-world assembly scenarios, (2) capture the comprehensive assembly knowledge, (3) follow a consistent annotation protocol that aligns both human and robot assembly comprehension. However, existing datasets fall short in meeting all these requirements.

First, the assembled products in existing datasets are either highly application-specific [4–9] or lack representative assembly parts and tools [5–7, 9]. Second, many datasets did not design assembly tasks to foster the emergence of natural behaviors (e.g., varying efficiency, alternative routes, pauses and errors) during procedural knowledge acquisition. Third, thorough understanding of nuanced assembly knowledge is challenging with existing datasets as they often do not to annotate subjects, objects, tools and their interactions in a systematic approach.

Therefore, we introduce HA-ViD: a human assembly video dataset recording people assembling the Generic Assembly Box (GAB, see Figure 1). We benchmark on four foundational video understanding

---

[*]These authors contributed equally to this work.
[†]Corresponding author.

tasks: action recognition, action segmentation, object detection and multi-object tracking (MOT), and analyze their performance and the further reasoning steps for comprehending application-oriented knowledge. HA-ViD features three novel aspects:

- **Representative industrial assembly scenarios**: GAB includes 35 standard and non-standard parts frequently used in real-world industrial assembly scenarios and requires 4 standard tools to assemble it. The assembly tasks are arranged onto 3 plates featuring different task precedence and collaboration requirements to promote the emergence of two-handed collaboration and parallel tasks. Compared to existing assembly video datasets, GAB is more representative of generic industrial assembly scenarios (see Table 1).

- **Natural procedural knowledge acquisition process**: Progressive observation, thought and practice process (shown as varying efficiency, alternative assembly routes, pauses, and errors) in acquiring and applying complex procedural assembly knowledge is captured via the designed three-stage progressive assembly setup (see Figure 1). This design allows in-depth understanding of the human cognition process, where existing datasets are limited (see Table 1).

- **Consistent human-robot shared annotations**: We designed a consistent fine-grained hierarchical task/action annotation protocol following a Human-Robot Shared Assembly Taxonomy (HR-SAT[1] , to be introduced in Section 2.3). Using this protocol, we, (1) granulate action annotations to subject, action verb, manipulated object, target object, and tool; (2) provide two-handed collaboration status annotations; and (3) annotate human pauses and errors. Such detailed annotation embeds more knowledge sources for diverse understanding of application-oriented knowledge (see Table 1).

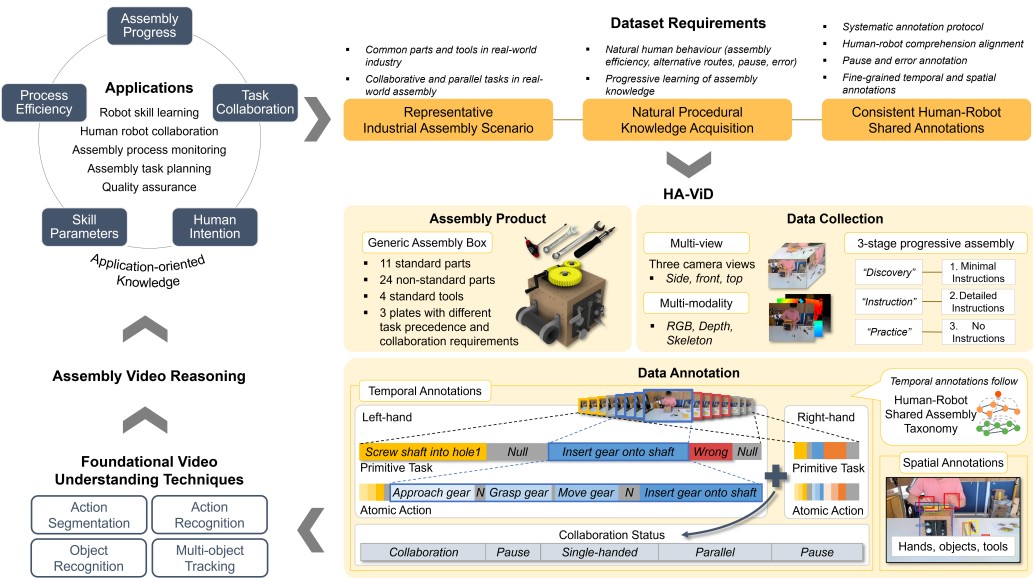

Figure 1: HA-ViD, a dataset designed for industrial applications, represents real-world assembly scenarios, and captures the process of acquiring procedural knowledge. The consistent annotation follows a human-robot shared taxonomy. The dataset features 3222 multi-view and multi-modalities videos, 1.5M frames, 96K temporal labels and 2M spatial labels.

## 2    Dataset

In this section, we present the process of building HA-ViD and provide essential statistics.

---

[1]HR-SAT, developed by the same authors, is a hierarchical assembly task representation schema that both humans and robots can comprehend. See details via: `https://iai-hrc.github.io/hr-sat`

Table 1: Comparison between HA-ViD and other assembly video datasets.

| Dataset | Assembled product | Natural procedural knowledge aquisition process | | | | Consistent human-robot shared assembly taxonomy | | | | | | Two-handed collaboration status |
| --- | --- | --- | --- | --- | --- | --- | --- | --- | --- | --- | --- | --- |
| | | Varying assembly efficiency | Alternative route | Pause | Error | Subject | Action verb | Manipulated object | Target object | Tool | Two-hand | |
| Wooden box [8] | Wooden box | × | × | × | × | × | ✓ | × | × | × | × | × |
| IKEA-FA [7] | Furniture | × | ✓ | ✓ | × | × | ✓ | ✓ | × | × | × | × |
| MECCANO [9] | Toy motorbike | × | ✓ | × | × | × | ✓ | ✓ | × | × | × | × |
| IKEA ASM [5] | Furniture | × | ✓ | ✓ | × | × | ✓ | ✓ | × | × | × | × |
| Assembly101 [6] | Toy cars | × | ✓ | × | ✓ | × | ✓ | ✓ | × | ✓ | × | × |
| HA4M [4] | Epicyclic Gear Train | × | ✓ | ✓ | × | × | ✓ | ✓ | × | × | × | × |
| HA-ViD (ours) | Generic assembly box | ✓ | ✓ | ✓ | ✓ | ✓ | ✓ | ✓ | ✓ | ✓ | ✓ | ✓ |

## 2.1 Generic Assembly Box

To ensure the dataset can represent real-world industrial assembly scenarios, we designed the GAB shown in Figure 1.

First, GAB[2] is a 250×250×250mm box including 11 standard and 24 non-standard parts frequently used in real-world industrial assembly. Four standard tools are required for assembling GAB. The box design also allows participants to naturally perform tasks on a top or side-facing plate, closer to the flexible setups of real-world assembly.

Second, GAB consists of three plates featuring different task precedence and collaboration requirements. Figure 2 shows the subject-agnostic task precedence graphs (SA-TPG) for the three plates with different precedence constraints. These different task precedence graphs provide contextual links between actions, enabling situational reasoning with different complexities. The cylinder plate also has more collaboration tasks, posing greater challenges for understanding collaborative assembly tasks. Gear and cylinder plates contain parts that become hidden after assembly, e.g., spacers under the gears. This introduces additional complexities for understanding assembly status.

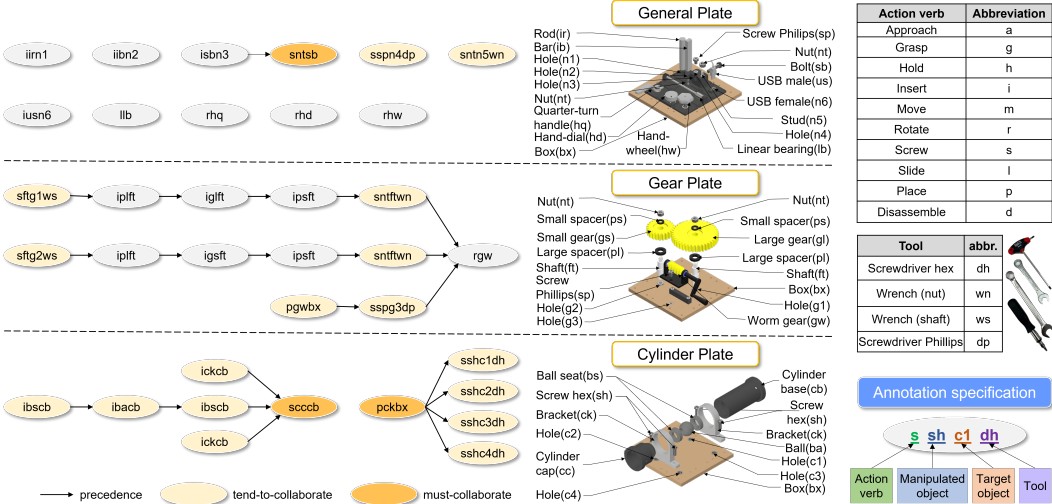

Figure 2: Subject-agnostic task precedence graphs for three plates and annotation specification. "must-collaborate" denotes the task requires two-handed collaboration, and "tend-to-collaborate" denotes the task that tend to need two hands.

### 2.1.1 Dataset Collection

Data was collected on three Azure Kinect RGB+D cameras mounted to an assembly workbench facing the participant from left, front and top views (see Figure 4). Videos were recorded at 1280×720 RGB resolution and 512×512 depth resolution under lab lighting and natural lighting conditions. 30 participants (15 male, 15 female) assembled each plate 11 to 12 times during a 2-hour session.

---

[2]Find GAB CAD files at: `https://iai-hrc.github.io/ha-vid`

To capture the progression of human procedural knowledge [10] acquisition and behaviors (e.g., varying efficiency, alternative routes, pause, and errors) during learning, a three-stage progressive assembly setup is designed. Inspired by discovery learning [11], we design the three stages as[3]: *Discovery* – participants are given minimal exploded view instructions of each plate; *Instruction* – participants are given detailed step-by-step instructions of each plate; *Practice* – participants are asked to complete the task without instruction.

The first stage encourages participants to explore assembly knowledge to reach a goal, the second stage provides targeted instruction to deepen participants' understanding, and the last stage encourages participants to reinforce their learning via practicing. During *Instruction* and *Practice* stages, the participants were asked to perform the assembly with the plate both facing upwards and sideways.

### 2.1.2 Dataset Annotations

We provide temporal and spatial annotations to capture rich assembly knowledge shown in Figure 1.

To enable human-robot assembly knowledge transfer, our annotations follow the HR-SAT (shown in Figure 3). According to the HR-SAT, an assembly task can be decomposed into primitive tasks and further into atomic actions. Each primitive task and atomic action contain five description elements: *subject*, *action verb*, *manipulated object*, *target object* and *tool*. Primitive tasks annotations describe a functional change of the manipulated object, such as inserting a gear on a shaft or screwing a nut onto a bolt. Atomic actions describe an interaction change between the subject and manipulated object such as a hand grasping the screw or moving the screw. HR-SAT ensures the transferability, adaptability, and consistency of annotations. It can be used to transform annotations from other datasets into our designated description, or annotate new data to extend HA-ViD.

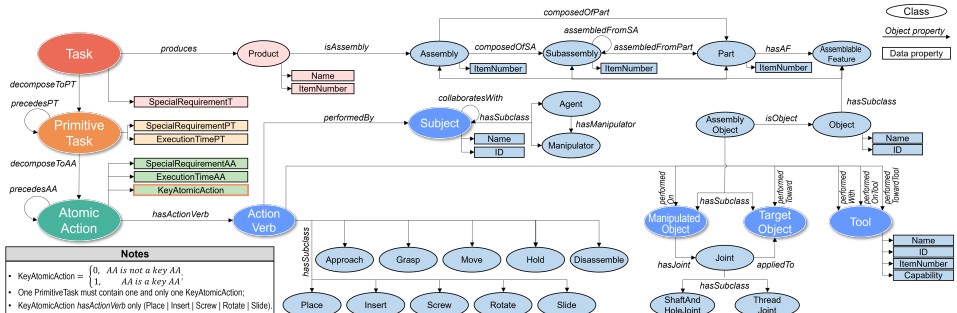

Figure 3: Human-robot shared assembly taxonomy (HR-SAT) schema. We tailored the original taxonomy by removing information that cannot be annotated from videos and incorporating a *Disassemble* action verb to describe human error-and-correction process. We provide textual annotations (see Figure 2) following the typical input formats of current video understanding algorithms. We also offer SA-TPGs as knowledge graphs [4] in RDF/XML format following the HR-SAT schema to enable advanced assembly knowledge reasoning with enhanced relationship information.

We annotate human pause and error as *null* and *wrong* respectively to enable research on understanding assembly efficiency and learning progression. Our annotations treat each hand as a separate subject. Primitive tasks and atomic actions are labeled for each hand to support multi-subject collaboration related research. Alongside the primitive task annotations, we annotate the two-handed collaboration status as: *collaboration*, when both hand work together on the same task; *parallel*, when each hand is working on a different task; *single-handed*, when only one hand is performing the task while the other hand pauses; and *pause*, when neither hand is performing any task.

For spatial annotations, we use CVAT[5], a video annotation tool, to manually label bounding boxes for subjects, objects, and tools frame-by-frame. Furthermore, we treat important assemblable features

---

[3]The instruction files can be found at `https://iai-hrc.github.io/ha-vid`. The detailed instructions were developed following HR-SAT to align assembly instructions with our annotations.

[4]The ST-TPGs files can be downloaded at: https://iai-hrc.github.io/hr-sat

[5]`https://www.cvat.ai/`

(such as holes, stud and USB female) as objects, to enable finer-grained assembly knowledge understanding.

## 2.2 Statistics

In total, we collected 3222 videos with side, front and top camera views. Each video contains one task – the process of assembling one plate. Our dataset contains 86.9 hours of footage, totaling over 1.5 million frames with an average of 1 min 37 sec per video (1456 frames). To ensure annotation quality, we manually labeled temporal annotations for 609 plate assembly videos and spatial annotations for over 144K frames. The selected videos for labeling collectively capture the dataset diversity by including videos of different participants, lightings, instructions and camera views.

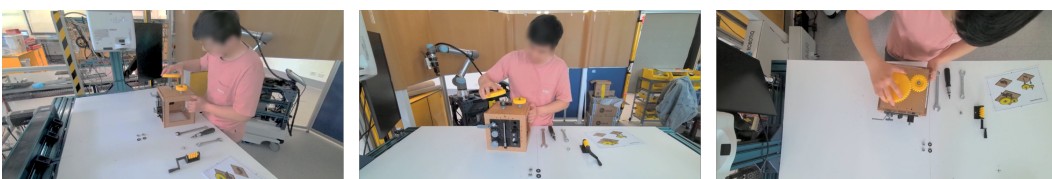

Figure 4: Side, front and top camera views of the workbench.

Overall, our dataset (in Fig. 5) contains 18831 primitive tasks across 75 classes, 63864 atomic actions across 219 classes, and close to 2M instances of subjects, objects and tools across 42 classes. Our dataset shows potential for facilitating small object detection research as 46.6% of the annotations are of small objects.

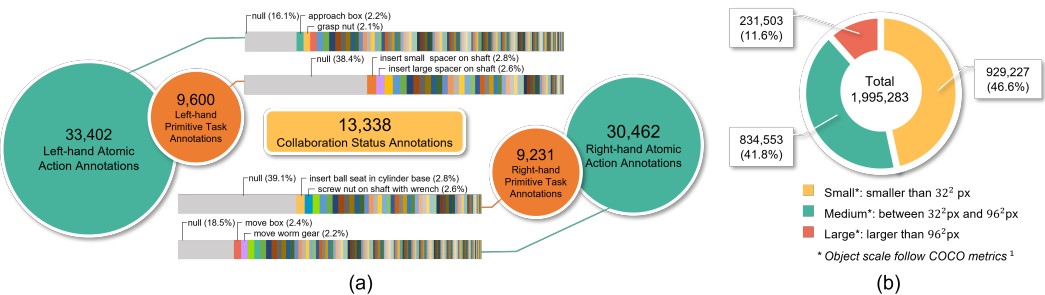

Figure 5: Temporal and spatial annotation statistics. (a) Total number of temporal annotations and annotation distributions, categorized by hands. The three head classes of primitive tasks and atomic actions are shown. (b) Total number of spatial annotations categorized into COCO object scale.

Our temporal annotations can be used to understand the learning progression and efficiency of participants over the designed three-stage progressive assembly setup. The combined annotation of *wrong* primitive task, *pause* collaboration status and total frames can indicate features such as errors, observation patterns and task completion time for each participant. Our dataset captures the natural progress of procedural knowledge acquisition, as indicated by the overall reduction in task completion time and pause time from stage 1 to 3, as well as the significant reduction in errors (see Figure 6). The *wrong* and *pause* annotations enable research on understanding varying efficiency between participants.

By annotating the collaboration status and designing three assembly plates with different task precedence and collaboration requirements, HA-ViD captures the two-handed collaborative and parallel tasks commonly featured in real-world assembly, shown in Figure 7. Overall, 49.6% of the annotated frames consist of collaborative or parallel tasks. The high percentage of two-handed tasks enables research in understanding the collaboration patterns of complex assembly tasks.

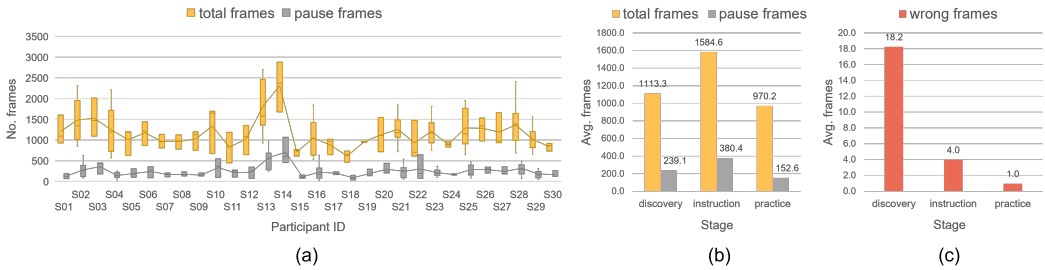

Figure 6: Annotation statistics of total frames, pause frames, and wrong frames. (a) Total frames and pause frames distribution by participant. (b) Average total frames and pause frames per task in each progressive assembly stage. (c) Average wrong frames per task in each progressive assembly stage.

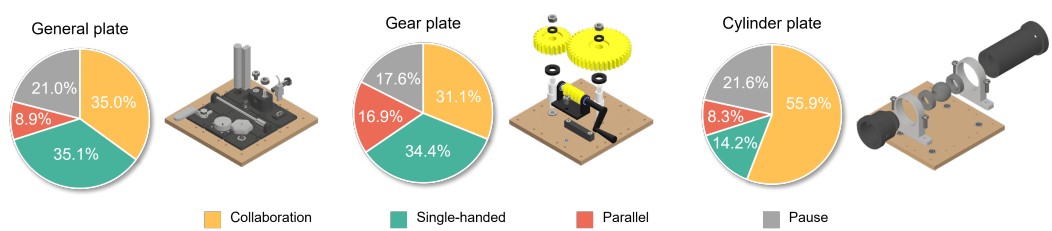

Figure 7: Percentage distribution of each collaboration status annotation for each assembly plate.

# 3 Benchmark Experiments

We benchmark SOTA methods for four foundational techniques for assembly knowledge understanding, i.e., action recognition, action segmentation, object detection, and MOT. Due to the page limit, we highlight key results and findings in this section, and present implementation details, more results and discussions in the Supplementary document.

## 3.1 Action Recognition, Action Segmentation, Object Detection and MOT

**Action recognition** is to classify a sequence of video frames into an action category. We split 123 out of 609 temporally labeled videos to be the testset, and the rest as trainset. We benchmark seven action recognition methods from three categories: 2D models (TSM [12], TimeSFormer [13]), 3D models (I3D [14], MVITv2 [15], UniFormerV2 [16]), and skeleton-based methods (ST-GCN [17], ST-GCN++ [18]) and report the Top-1 accuracy and Top-5 accuracy in Table 2.

**Action segmentation** is to temporally locate and recognize human action segments in untrimmed videos [19]. Under the same train/test split as action recognition, we benchmark four action segmentation methods, MS-TCN [20], DTGRM [19], BCN [21], and C2F-TCN [22], and report the frame-wise accuracy (Acc), segmental edit distance (Edit) and segmental F1 score at overlapping thresholds of 10% in Table 3.

**Object detection** is to detect all instances of objects from known classes [23]. We split 18.4K out of 144K spatially labeled frames to be the testset, and the rest as trainset. We benchmark classical two-stage method FasterRCNN [24], one-stage method Yolov5 [25], and the SOTA end-to-end Transformer-based method DINO [26] with different backbone networks, and report the parameter size (Params), average precision (AP), AP under different IoU thresholds (50% and 75%), and AP under different object scales (small, medium and large) in Table 4.

**MOT** aims at locating multiple objects, maintaining their identities, and yielding their individual trajectories given an input video [27]. We benchmark SORT [28] and ByteTrack [29] on the detection results of DINO and ground truth annotations (test split of object detection), respectively. We report the average multi-object tracking accuracy (MOTA), ID F1 score (IDF1), false positive (FP), false negative (FN), and ID switch (IDS) over the videos in our testing dataset in Table 5.

Table 2: Baselines of action recognition. Average results over three views are reported here and more detailed results can be found in the Supplementary document.

| Method | View | Primitive Task | | | | Atomic Action | | | |
| | | Left-Hand | | Right-Hand | | Left-Hand | | Right-Hand | |
| | | Top-1 | Top-5 | Top-1 | Top-5 | Top-1 | Top-5 | Top-1 | Top-5 |
|---|---|---|---|---|---|---|---|---|---|
| TSM [12] | Average | 61.0 | 88.5 | 58.6 | 87.9 | 39.6 | 69.4 | 37.0 | 67.2 |
| TimeSFormer [13] | Average | 52.1 | 85.4 | 51.8 | 84.4 | 37.6 | 68.8 | 34.6 | 66.1 |
| I3D(rgb+flow) [14] | Average | 47.7 | 71.5 | 52.9 | 85.1 | 43.0 | 75.0 | 40.5 | 72.9 |
| MViTv2 [15] | Average | 61.5 | 86.3 | 58.7 | 84.1 | 48.4 | 76.5 | 42.9 | 71.2 |
| UniFormerV2 [16] | Average | **62.4** | **89.7** | **61.4** | **89.9** | **48.9** | **80.9** | **44.6** | **77.6** |
| ST-GCN [17] | Average | 39.5 | 60.2 | 38.7 | 55.2 | 20.3 | 44.4 | 19.7 | 40.6 |
| ST-GCN++ [18] | Average | 38.8 | 58.0 | 37.5 | 56.7 | 19.0 | 41.3 | 16.7 | 36.1 |

Table 3: Baselines of action segmentation. Average results over three views are reported here and detailed results can be found in the Supplementary document.

| Method | View | Primitive task | | | | | | Atomic Action | | | | | |
| | | Left hand | | | Right hand | | | Left hand | | | Right hand | | |
| | | F1 | Edit | Acc | F1 | Edit | Acc | F1 | Edit | Acc | F1 | Edit | Acc |
|---|---|---|---|---|---|---|---|---|---|---|---|---|---|
| MS-TCN [20] | Avg. | 36.6 | 37.5 | 40.2 | 34.7 | 34.8 | 39.3 | **35.1** | 32.5 | **40.9** | **31.2** | 32.2 | **34.6** |
| DTGRM [19] | Avg. | 39.1 | 37.5 | 40.2 | 37.8 | 37.3 | 39.7 | 34.3 | **32.6** | 39.8 | 29.8 | 29.3 | 33.1 |
| BCN [21] | Avg. | **43.7** | **41.4** | **44.1** | **41.3** | **38** | **43.4** | 18.4 | 15.9 | 39.7 | 22.3 | 20.1 | **34.6** |
| C2F-TCN [22] | Avg. | 22.6 | 22.0 | 39.5 | 22.5 | 21.9 | 39 | 20.5 | 19.2 | 37.6 | 17.4 | 17.4 | 31.8 |

Table 4: Baselines of object detection.

| Method | Backbone | Params | AP | AP50 | AP75 | AP-s | AP-m | AP-l |
|---|---|---|---|---|---|---|---|---|
| | ResNet50 | 41.6M | 21.7 | 32.6 | 24.4 | 13.0 | 37.4 | 40.6 |
| Faster-RCNN [24] | ResNet101 | 60.6M | 20.9 | 31.1 | 23.9 | 12.3 | **37.9** | 43.1 |
| | ResNext101 | 99.5M | 22.2 | 31.6 | 25.7 | 15.0 | 36.2 | 46.2 |
| YOLOv5-s [25] | DarkNet | 7.1M | 10.2 | 14.1 | 10.9 | 0.7 | 18.8 | 46.8 |
| YOLOv5-l [25] | DarkNet | 46.4M | 12.9 | 17.3 | 14.0 | 1.0 | 28.8 | **59.8** |
| DINO [26] | Swin-L | **218M** | **35.5** | **54.5** | **37.7** | **27.4** | 36.4 | 59.2 |

Table 5: MOT results on object detection results and ground truth object bounding boxes.

| Method | bboxes | MOTA | IDF1 | FP | FN | IDS |
|---|---|---|---|---|---|---|
| SORT [28] | dets | **20.4%** | 27.1% | **737.8** | 9212.3 | **29** |
| | gt | 94.5% | **69.1%** | 223.9 | 408.1 | 54.8 |
| ByteTrack [29] | dets | 20.0% | **41.1%** | 5175.3 | **4678.3** | 87.2 |
| | gt | **98.5%** | 67.5% | **32.4** | **32.5** | 121.6 |

The baseline results show that our dataset presents great challenges on the four foundational video understanding tasks compared with other datasets. For example, BCN has 70.4% accuracy on Breakfast [30], MViTv2 has 86.1% Top-1 accuracy on Kinetics-400 [31], DINO has 63.3% AP on COCO test-dev [32], and ByteTrack has 77.8% MOTA on MOT20 [33].

Compared to the above baseline results, we are more concerned with whether existing video understanding methods can effectively comprehend the application-oriented knowledge (shown in Figure 1). Therefore, in Sections 3.2 to 3.5, we further analyze the performance and limitation of the foundational tasks on comprehending application-oriented knowledge, discuss the required assembly video reasoning tasks, and highlight the potential research directions.

## 3.2 Assembly progress

**Insight #1: Assembly progress understanding could focus on compositional action understanding and leveraging prior domain knowledge.** Basic assembly progress understanding requires real-time action (action verb + interacted objects and tools) recognition, and compare the action history with the predefined assembly plan (represented in a task graph). After further analysis of the sub-optimal action recognition performance in Table 2, we found recognizing the interacting objects and tools are more challenging than recognizing the action verbs, (as shown in Table 6).

Table 6: Recall of action verb, manipulated object, target object, and tool recognition, via MVITv2.

|  | Action verb | Manipulated Object | Target Object | Tool |
|---|---|---|---|---|
| Primitive Task | 71.1% | 60.4% | 57.1% | 60.8% |
| Atomic Action | 67.6% | 50.9% | 53.5% | 55.0% |

Deeper assembly progress understanding requires real-time reasoning on human-object interaction dynamics, future operations, and their operation times. Taking a step to address this need, we benchmark FUTR [34] – a SOTA long-term action anticipation method, and report the mean over classes (MoC) accuracy as per [35] in Table 7. Similar to Table 6, a higher accuracy of action verb anticipation can also be observed in Table 7.

Table 7: Mean over classes accuracy of action (action verb + interacted objects and tools) anticipation, action verb anticipation and manipulated object anticipation. Following the problem setting in [34]: for a video with $T$ frames, the first $\alpha T$ frames are observed and anticipate the next $\beta T$ frames.

|  |  | $\beta(\alpha=0.2)$ | | | | $\beta(\alpha=0.3)$ | | | |
|---|---|---|---|---|---|---|---|---|---|
|  |  | 0.1 | 0.2 | 0.3 | 0.5 | 0.1 | 0.2 | 0.3 | 0.5 |
| Action anticipation | Primitive task | 7.4 | 6.1 | 4.7 | 4.4 | 12.3 | 9.2 | 6.5 | 5.6 |
|  | Atomic action | 2.2 | 1.8 | 1.6 | 1.4 | 4.7 | 2.7 | 2.2 | 1.8 |
| Action verb anticipation | Primitive task | **24.2** | **23.5** | **18.2** | **17.6** | **32.8** | **18.6** | **16.6** | **16.0** |
|  | Atomic action | **14.6** | **12.0** | **10.7** | **10.4** | **15.4** | **13.0** | **11.7** | **10.2** |
| Manipulated object anticipation | Primitive task | 13.5 | 10.0 | 9.1 | 9.6 | 17.0 | 12.9 | 10.7 | 9.1 |
|  | Atomic action | 9.0 | 9.1 | 8.7 | 8.1 | 13.5 | 10.7 | 9.8 | 8.3 |

Based on the observation from Table 6 and 7, a promising research direction could be recognizing and anticipating action verbs, objects, and tools compositionally and leveraging prior domain knowledge (such as task precedence and probabilistic correlation between action verbs, objects, and tools) to precisely track and predict the assembly progress. With defined task precedence graphs and rich list of annotated action verb/object/tool pairs, HA-ViD enables research on this aspect.

**Insight #2: Assembly action segmentation should focus on addressing under-segmentation issues and improving segment-wise sequence accuracy.** Assembly progress tracking requires obtaining the accurate number of action segments and their sequence. For obtaining the accurate number of action segments from a given video, previous action segmentation algorithms [19–21] focused on addressing over-segmentation issues, but lack metrics for quantifying under/over-segmentation. Therefore, we propose segmentation adequacy (SA) to fill this gap. Considering the predicted segments as $s_{\mathrm{pred}} = \{s'_1, s'_2, \ldots, s'_F\}$ and ground truth segments as $s_{\mathrm{gt}} = \{s_1, s_2, \ldots, s_N\}$, where $F$ and $N$ are the number of segments for a given video, $\mathrm{SA} = \tanh\left(\frac{2(F-N)}{F+N}\right)$. Table 8 reveals the significant under-segmentation issues observed from our dataset, which is a potentially important issue to be addressed for assembly action understanding. Our proposed SA metric can offer evaluation support, and even assist in designing the loss function given its use of the hyperbolic tangent function.

Table 8: Comparison between our dataset and others on segmentation adequacy. We calculated the average number of ground truth segments ($N$), predicted segments ($F$), and segment adequacy ($SA$) of the videos in the testing datasets of ours and others. The predicted results are from BCN.

| Dataset |  | $N$ | $F$ | $SA$ |
|---|---|---|---|---|
| HA-ViD(ours) | Primitive task | 14.9 | 8.3 | -0.47 |
|  | Atomic action | 51.2 | 11.5 | -0.82 |
| Breakfast |  | 6 | 6.8 | -0.12 |
| GTEA |  | 32.5 | 32.9 | -0.03 |

For segment-wise sequence accuracy, the low Edit value in Table 3 suggests further research efforts are required. Compared with Breakfast [30] (66.2% Edit score with BCN), our dataset presents greater challenges.

## 3.3 Process Efficiency

Understanding process efficiency is essential for real-world industry. It requires video understanding methods to be capable of identifying human pause and error via reasoning of the contextual scene and human-object interaction. HA-ViD supports this research by providing *null* and *wrong* labels.

**Insight #3: *Null* action understanding requires efforts on addressing imbalanced class distribution.** Table 9 shows the recall and precision of action recognition and action segmentation for *null* actions. We suspect the high recall and low precision is caused by the imbalanced class distribution, as null is the largest head class (see Figure 5). To address the class imbalance problem, we randomly over-sample the minority classes (classes with sample size below a threshold) to reach the threshold, and report the action recognition accurary of UniFormerV2 on the over-sampled dataset in Table 10. Here, the threshold is set to 300 and more details can be found in the Supplementary.

Table 9: Recall and precision of *null* recognition and segmentation. Action recognition results are from MVITv2 and action segmentation results are from BCN.

|  |  | Recall | Precision |
|---|---|---|---|
| Recognition | Primitive Task | 90.8% | 65.1% |
|  | Atomic Action | 81.5% | 39.1% |
| Segmentation | Primitive Task | 80.9 | 45.1% |
|  | Atomic Action | 84.6% | 37.5% |

Table 10: The action recognition accuracy of UniFormerV2 on the over-sampled dataset.

| Method | View | Primitive Task | | | | Atomic Action | | | |
|---|---|---|---|---|---|---|---|---|---|
| | | Left-hand | | Right-hand | | Left-hand | | Right-hand | |
| | | Top-1 | Top-5 | Top-1 | Top-5 | Top-1 | Top-5 | Top-1 | Top-5 |
| UniFormerV2 | Average | 62.4 | **89.7** | 61.4 | **89.9** | 48.9 | 80.9 | 44.6 | 77.6 |
| UniFormerV2 (over-sampling) | Average | **67.3** | 89.2 | **66.2** | 89.1 | **52.0** | **81.2** | **48.8** | **78.1** |

**Insight #4: New research from *wrong* action annotations.** *Wrong* actions refer to assembly actions (primitive task level) occurred at the wrong position or order. Our annotated wrong actions can initiate three avenues for research. First, the pattern of wrong actions in different participants across the three-stage progressive assembly can provide insights into human learning progression and performance. Second, investigating the wrong action and the actions leading up to it can help identify assembly errors, contributing to quality assurance. Third, analyzing the actions undertaken after the wrong action can provide insights into how humans resolve errors and re-plan the assembly sequence.

## 3.4   Task Collaboration

Understanding the states, patterns, and dynamics of two-handed collaboration during the assembly process is essential for applications, such as ergonomics analysis, collaborative task planning, and human-robot collaboration design. HA-ViD can support research in this aspect via providing spatial annotations, two-hand separated temporal annotations and collaboration status annotations.

**Insight #5: New research on understanding parallel tasks from both hands.** Table 11 shows that both action recognition and segmentation have lowest performance on parallel tasks during assembly. One possible reason is that the foundational video understanding methods rely on global features of each image, and do not explicitly detect and track the action of each hand. This calls for new methods to independently track both hands and recognize their actions through local features. Recent research on human-object interaction detection in videos [36, 37] could offer valuable insights.

Table 11: Recall of two-handed primitive task recognition and segmentation in four collaboration status. Action recognition results are from MVITv2 and action segmentation results are from BCN.

| | Action recognition results | | | | Action segmentation results | | | |
|---|---|---|---|---|---|---|---|---|
| | Collaboration | Parallel | Single-handed | Pause | Collaboration | Parallel | Single-handed | Pause |
| Left hand | 52.5% | 39.7% | 54.2% | 92.4% | 32.1% | 15.4% | 18.5% | 85.5% |
| Right hand | 46.1% | 30.5% | 50.7% | 93.3% | 35.0% | 24.2% | 17.2% | 82.9% |

## 3.5   Skill Parameters and Human Intention

Understanding skill parameters and human intentions from videos is essential for robot skill learning and human-robot collaboration (HRC) [38, 39].

Typically, skill parameters vary depending on the specific application. However, there are certain skill parameters that are commonly used, including trajectory, object pose, force, and torque [40, 41].

While videos cannot capture force and torque directly, our dataset offers spatial annotations that enable tracking the trajectory of each object. Additionally, the object pose can be inferred from our dataset via pose estimation methods. Therefore, HA-ViD can support basic research in this direction. To acquire more detailed and accurate skill parameters, assembly process comprehension must extend to the 3D space, utilizing 3D reasoning and 3D hand-object interaction estimation techniques to precisely track the temporal changes in hand poses and objects. For human-robot skill transfer, the learned assembly skill parameters can be sourced as input to robot learning environments.

Understanding human intention in HRC refers to a combination of trajectory prediction, action anticipation and task goal understanding [42]. Our spatial annotations provide trajectory information, SA-TPGs present action sequence constraints, and GAB CAD files offer the final task goals. Therefore, HA-ViD can enhance the research in this aspect.

## 4 Discussion

As identified in Section 3, HA-ViD provides a basis for developing video understanding and reasoning techniques to derive application-oriented knowledge. Model design, results analysis, and knowledge reasoning pipeline development that are based on HA-ViD can accelerate the development of application-specific models. In addition, if a new dataset is required for a specific application, our HR-SAT-based annotation protocol can be employed to ensure the resulting target dataset is compatible with HA-ViD. The annotation alignment eases the process of adapting and deploying pre-trained models to new scenarios.

We acknowledge the limitation of HA-ViD on fully capturing the complexities and diversities of industrial assembly scenarios. Therefore, HA-ViD can be extended to include more products, assembly environments, and even different agents. Following our data collection and annotation protocol could ensure similar high-quality assembly video datasets that are compatible with HA-ViD.

Furthermore, we identify that HA-ViD could benefit from 3D and pixel-wise geometric annotations. This can facilitate the research into 3D hand-object interaction and 3D scene understanding, which is essential for assembly quality checking and robot skill learning. Therefore, future work could involve providing more refined annotations, such as hand poses, object poses and 3D key points, via additional sensors. We therefore created a dataset roadmap on our Github repository to outline improvement focuses and encourage collective efforts from the community to extend HA-ViD.

## 5 Conclusion

We present HA-ViD, a human assembly video dataset, to advance comprehensive assembly knowledge understanding toward real-world industrial applications. We designed the Generic Assembly Box to represent industrial assembly scenarios and a three-stage progressive learning setup to capture the natural process of human procedural knowledge acquisition. The dataset annotation follows the Human-Robot Shared Assembly Taxonomy. HA-ViD includes (1) multi-view, multi-modality data, fine-grained action annotations (subject, action verb, manipulated object, target object, and tool), (2) human pause and error annotations, and (3) collaboration status annotations to enable technological breakthroughs industrial application-oriented knowledge comprehension from videos.

We benchmarked strong baseline methods of action recognition, action segmentation, object detection and multi-object tracking, and analyzed their performance and the further reasoning steps for comprehending application-oriented knowledge in assembly progress, process efficiency, task collaboration, skill parameter and human intention. The results show that our dataset captures essential challenges for foundational video understanding tasks, and new methods need to be explored for application-oriented knowledge comprehension. We envision HA-ViD will open opportunities for advancing video understanding techniques to enable the futuristic ultra-intelligent industry.

## 6 Acknowledgements

This work was supported by The University of Auckland FRDF New Staff Research Fund (No. 3720540). Our gratitude extends to the participants and annotators of our dataset for their essential contributions. The Industrial AI Research group has provided invaluable feedback and advice, with

particular thanks to Benedict Liang for benchmarking assistance, and Saahil Chand for his annotations and insights.

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
