# *Supplementary Document* for HA-ViD: A Human Assembly Video Dataset for Comprehensive Assembly Knowledge Understanding

**Hao Zheng**[*]
hzhe951@aucklanduni.ac.nz

**Regina Lee**[*]
klee702@aucklanduni.ac.nz

**Yuqian Lu**[†]
yuqian.lu@auckland.ac.nz
Department of Mechanical and Mechatronics Engineering
The University of Auckland
https://iai-hrc.github.io/ha-vid

## 1   Overview

This supplementary document contains additional information about HA-ViD.

Section 2 further describes the process of building HA-ViD, including the design of the Generic Assembly Box, data collection, data annotation, and annotation statistics.

Section 3 presents the implementation details of our baselines, discusses the experimental results, and provides the licenses of the benchmarked algorithms.

Section 4 discusses the bias and societal impact of HA-ViD.

Section 5 presents the research ethics for HA-ViD.

Section 6 describes the datasheet for HA-ViD.

## 2   HA-ViD Construction

In this section, we further discuss the process of building HA-ViD. First, we introduce the design of the Generic Assembly Box. Second, we describe the three-stage data collection process. Third, we describe data annotation details. Finally, we present critical annotation statistics.

### 2.1   Generic Assembly Box Design

To ensure the dataset is representative of real-world industrial assembly scenarios, we designed the Generic Assembly Box (GAB), a 250×250×250mm box (see Figure 1), which consists of 11 standard parts and 25 non-standard parts and requires 4 standard tools during assembly (see Figure 2).

GAB has three assembly plates, including **General Plate**, **Gear Plate**, and **Cylinder Plate**, and three blank plates. The opposite face of each assembly plate is intentionally left blank to allow a different assembly orientation. Three assembly plates feature different design purposes.

**General Plate** (see Figure 3) was designed to capture action diversity. The general plate consists of 11 different parts. The parts used in this plate were designed to include the different directions,

---

[*]These authors contributed equally to this work.
[†]Corresponding author.

37th Conference on Neural Information Processing Systems (NeurIPS 2023) Track on Datasets and Benchmarks.

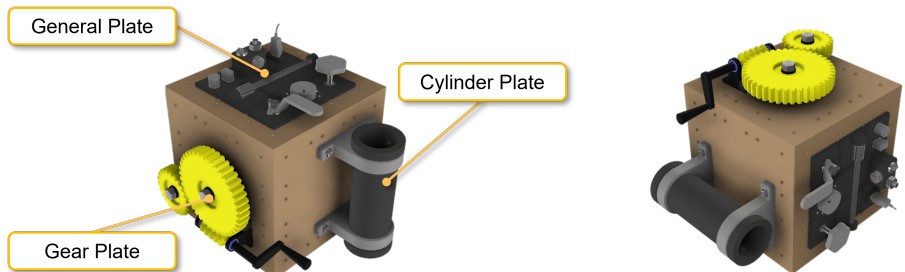

Figure 1: The fully assembled Generic Assembly Box is shown in two different orientations. Each plate can be assembled facing upwards or sideways.

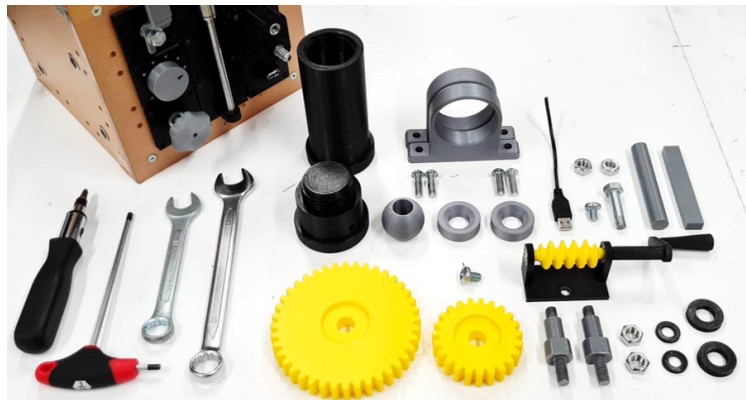

Figure 2: The Generic Assembly Box consists of 11 standard parts and 25 non-standard parts and requires 4 different standard tools during assembly.

shapes, and forces in which the common assembly actions can be performed. Since there is close to no precedence between assembling different parts, General Plate results in the most variety of possible assembly sequences.

**Gear Plate** (see Figure 4) was designed to capture parallel two-handed tasks, e.g., two hands inserting two spur gears at the same time. Gear Plate has three gear sub-systems: large gear, small gear, and worm gear, which mesh together to form a gear mechanism. The plate consists of 12 different parts. Gear Plate has a higher precedence constraint on assembly sequence than the general plate.

**Cylinder Plate** (see Figure 5) was designed to capture two-handed collaborative tasks, e.g., two hands collaborating on screwing the cylinder cap onto the cylinder base. Cylinder Plate requires assembling a cylinder subassembly and fastening it onto the plate. This plate consists of 11 parts. The parts were designed to represent assembling a subassembly where parts become fully occluded or partially constrained to another part (see the cylinder in Figure 5).

Table 1 shows a summary of the three assembly plates. The box can be easily replicated using standard components, laser cutting, and 3D printing. The CAD files and bill of material can be downloaded from our website[1].

Table 1: Summary of the three Generic Assembly Box plates.

| Plate | Design purpose | Precedence constraint | Two-handed collaboration | Standard Parts | Non-standard parts | Tools |
|---|---|---|---|---|---|---|
| General | Action and assembly sequence variety and minimal precedence | Minimal | Low | 4 | 7 | 2 |
| Gear | Parallel tasks and high precedence. | High | Medium | 3 | 9 | 3 |
| Cylinder | Collaboration tasks and high precedence. | High | High | 4 | 7 | 1 |

---

[1]https://iai-hrc.github.io/ha-vid

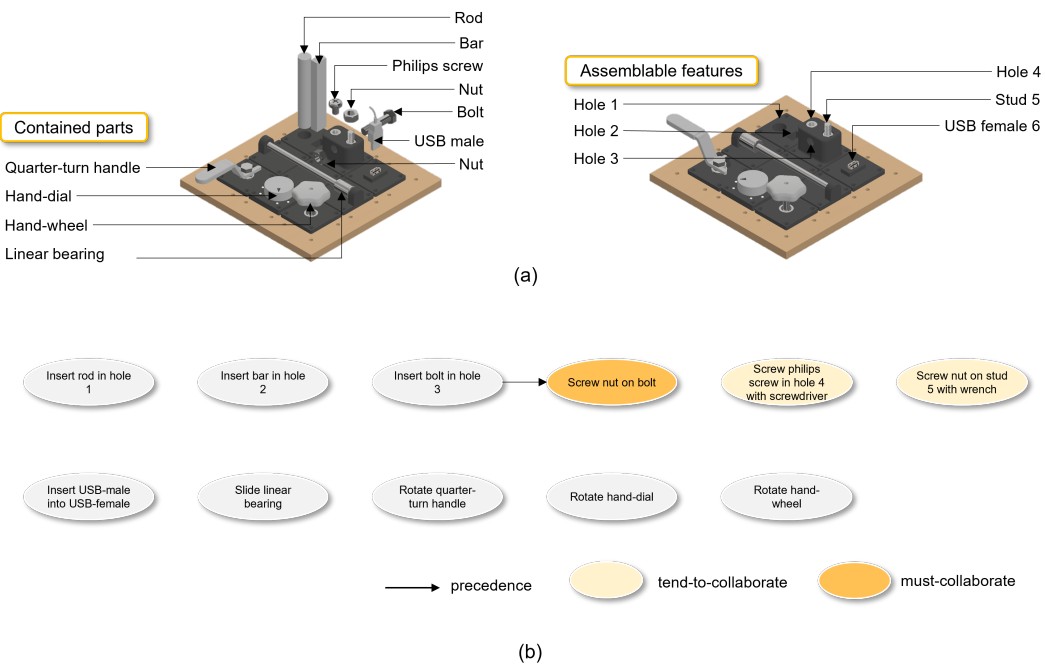

Figure 3: The general plate (a) the contained parts and assemblable features and (b) subject-agnostic task precedence graph where "must-collaborate" denotes the task requires two-handed collaboration, and "tend-to-collaborate" denotes the task that tend to need two hands. Different from general assembly datasets, we treat assemblable features, such as holes, stud and USB female, as objects, to enable finer-grained assembly knowledge understanding.

## 2.2 Data Collection

Data was collected under two lighting conditions with three Azure Kinect RGB+D cameras mounted to an assembly workbench. 30 participants (15 male, 15 female) were recruited for a 2-hour session to assemble the GAB. During the data collection session, participants were given a fully disassembled assembly box, assembly parts, tools, and instructions (see Figure 6).

To capture the natural progress of human procedural knowledge acquisition and behaviors (varying efficiency, alternative routes, pauses, and errors), we designed a three-stage progressive assembly setup:

*Discovery*: Participants were asked to assemble a plate twice following the minimal visual instructions (see Figure 7).

*Instruction*: Participants were asked to assemble a plate six times following the detailed step-by-step instructions (see Figure 8). Six different instruction versions were created, each presenting a different assembly sequence. Each participant was given three different instruction versions, where two attempts were completed following each instruction version. The three instruction versions given to one participant must contain assembling the plate facing both upwards and sideways.

*Practice*: After the first two stages, participants were asked to assemble a plate four times without any instructions. During this stage, participants performed two attempts of each plate facing upwards and two attempts of each plate facing sideways.

The instruction files are available on our website[2].

## 2.3 Data Annotation

To capture rich assembly knowledge, we provide temporal and spatial annotations.

---

[2]https://iai-hrc.github.io/ha-vid

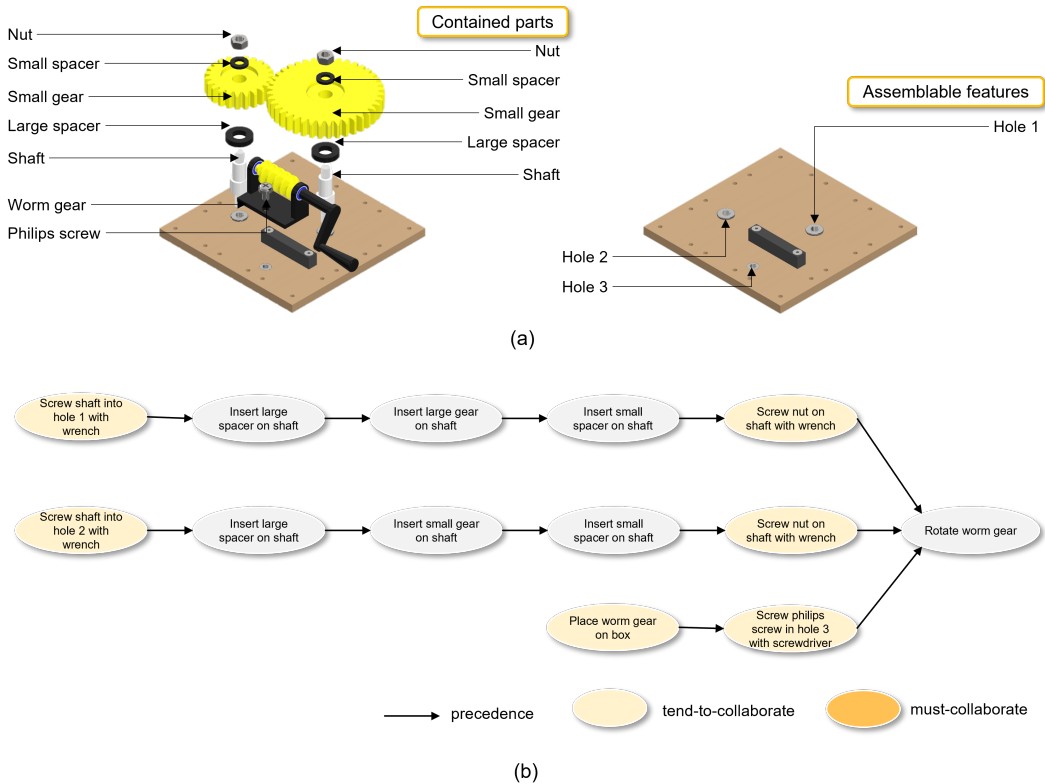

Figure 4: The gear plate (a) the contained parts and assemblable features and (b) subject-agnostic task precedence.

**Temporal Annotations**: In HR-SAT[3], an assembly task can be decomposed into a series of primitive tasks, and each primitive task can be further decomposed into a series of atomic actions. For both primitive task and atomic action, there are five fundamental description elements: *subject*, *action verb*, *manipulated object*, *target object*, and *tool* (see Figure 9). We follow HR-SAT to provide primitive task and atomic action annotations for the assembly processes recorded in the videos. To enable the research in two-handed collaboration task understanding, we defined the two hands of each participant as two separate subjects, and we annotated *action verb*, *manipulated object*, *target object*, and *tool* for each *subject*. For both primitive task and atomic action annotations, we follow the annotation specification shown in Figure 10.

**Spatial Annotations**: For spatial annotations, we use CVAT[4] to annotate the subjects (two hands), objects (manipulated object, target object), and tools via bounding boxes, shown in Figure 11.

## 2.4 Dataset Statistics

Overall, HA-ViD contains 1074 unique assembly processes captured from 3 camera views, providing 3222 videos in total. Each video contains one task, where a task is the assembly of one of the assembly plates of GAB. Table 2 below compares the quantitative statistics of HA-ViD with other existing datasets.

The dataset contains temporal annotations of 81 primitive task classes and 219 atomic action classes. The trainset and testset were split by subjects to balance data diversity. Figure 12 and Figure 13 show the class distributions of primitive task and atomic action annotations in the trainset and testset, respectively.

---

[3]Details for the definitions of primitive task and atomic action can be found at: https://iai-hrc.github.io/hr-sat
[4]https://www.cvat.ai/

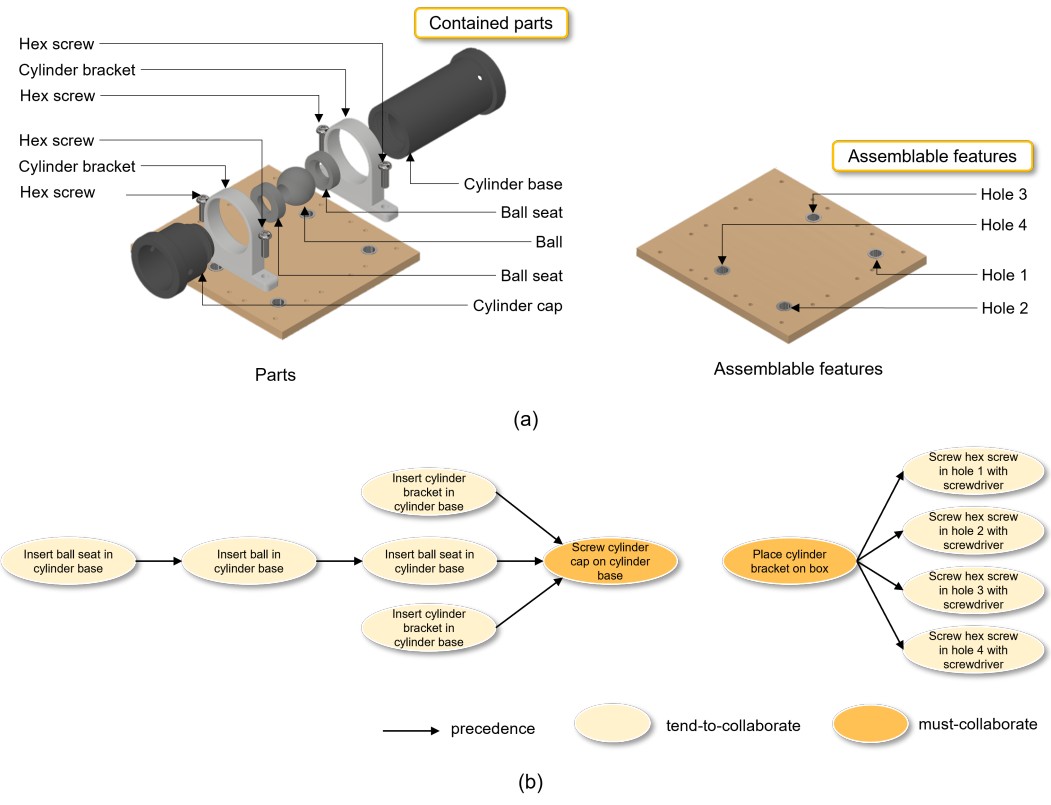

(a)

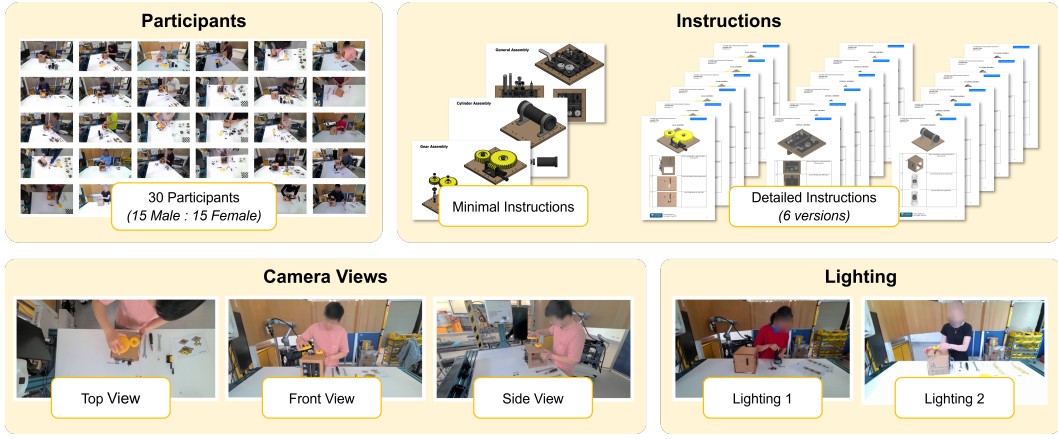

(b)

Figure 5: The cylinder plate (a) the contained parts and assemblable features and (b) subject-agnostic task precedence.

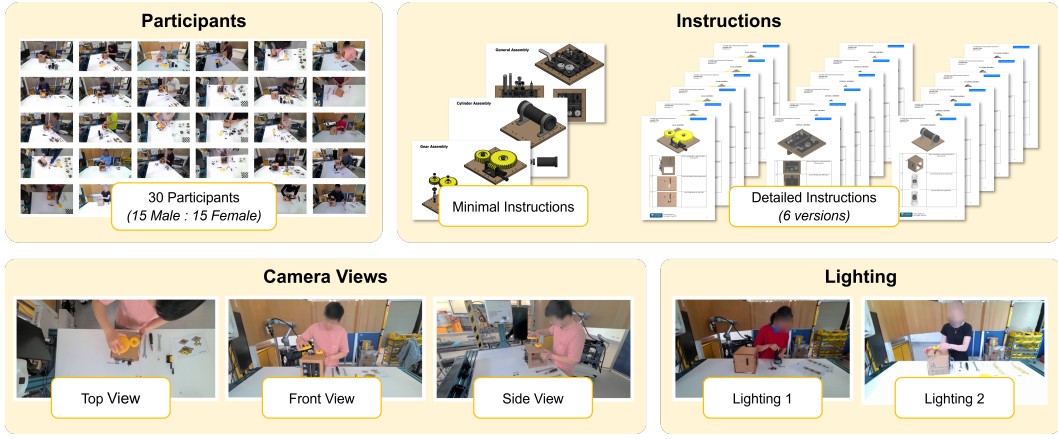

Figure 6: Summary of the data collection setup.

Table 2: Quantitative comparison between HA-ViD with other assembly video datasets.

| Dataset | Subject | Videos | Views | Unique sequences | Total duration (hr) | Average clip length (min) | Temporal label classes | Spatial label classes |
|---|---|---|---|---|---|---|---|---|
| Wooden box [1] | 17 | 124 | 2 | 62 | 13 | 12.8 | 9 | n/a |
| IKEA-FA [2] | 14 | 101 | 1 | 101 | n/a | 2 to 4 | 12 | n/a |
| MECCANO [3] | 20 | 20 | 1 | 20 | 6.9 | 21.1 | 61 | 20 |
| IKEA ASM [4] | 48 | 1113 | 3 | 371 | 11.7 | 1.9 | 33 | 9 |
| Assembly101 [5] | 53 | 4321 | 12 | 362 | 42.8 | 7.1 | 1380 fine 202 coarse | 90 |
| HA4M [6] | 41 | 217 | 1 | 217 | 5.9 | 1.6 | 13 | n/a |
| HA-ViD (ours) | 30 | 3222 | 3 | 1074 | 29 | 1.6 | 75 primitive tasks 219 atomic actions 4 collaboration status | 42 |

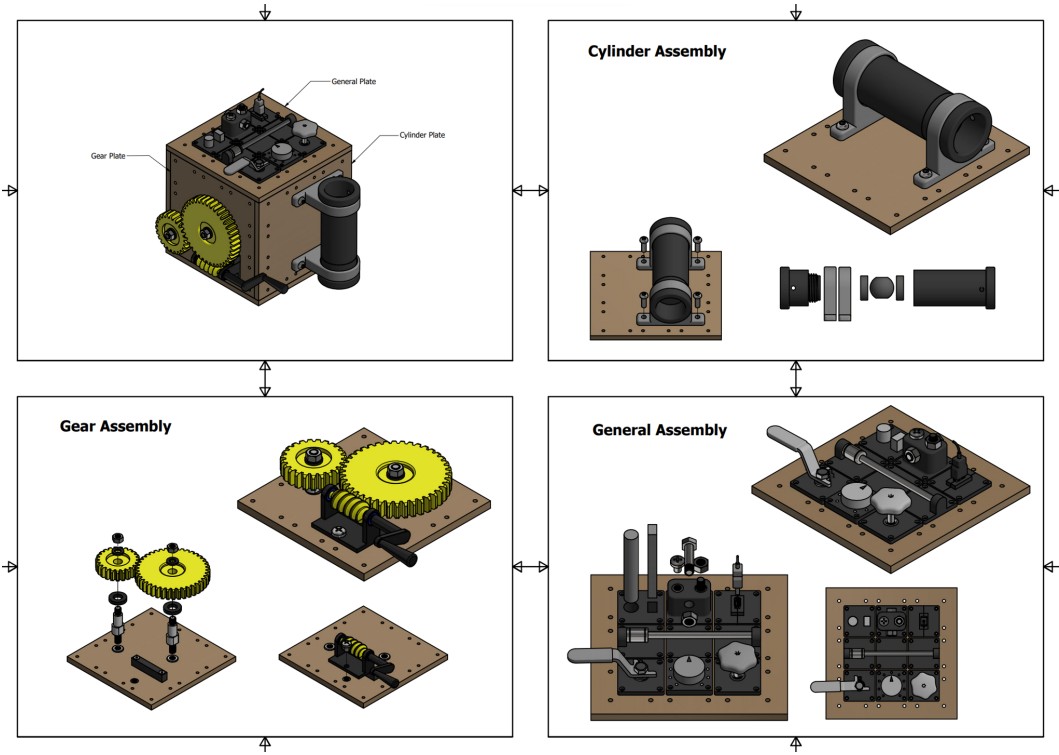

Figure 7: Minimal instruction pages.

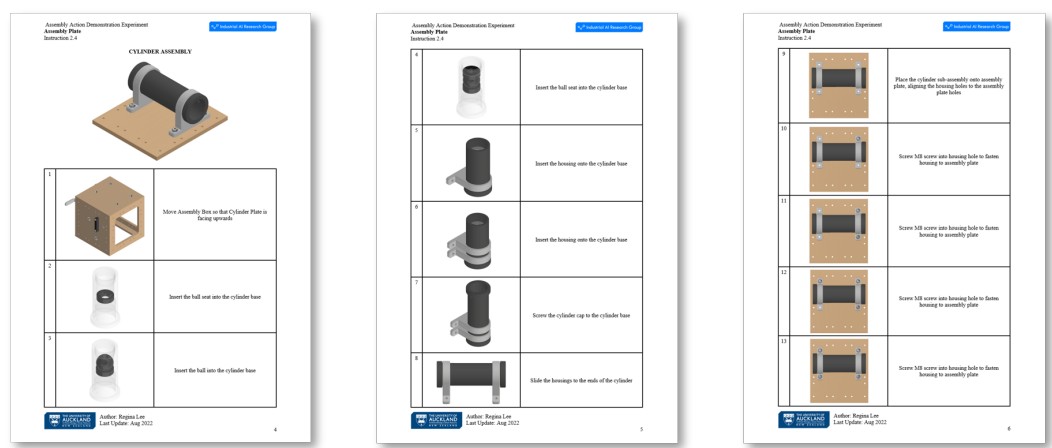

Figure 8: Example of the detailed instruction provided to participants for the cylinder assembly plate.

Overall, the dataset contains spatial annotations of 42 classes. The trainset and testset were split by subjects to balance data diversity. Figure 14 shows the class distributions of spatial annotation classes in the trainset and testset.

# 3   Experiment

In this section, we provide the implementation details of the baselines, the results unreleased in the main paper, further discussions on the results, and the licenses of the benchmarked algorithms.

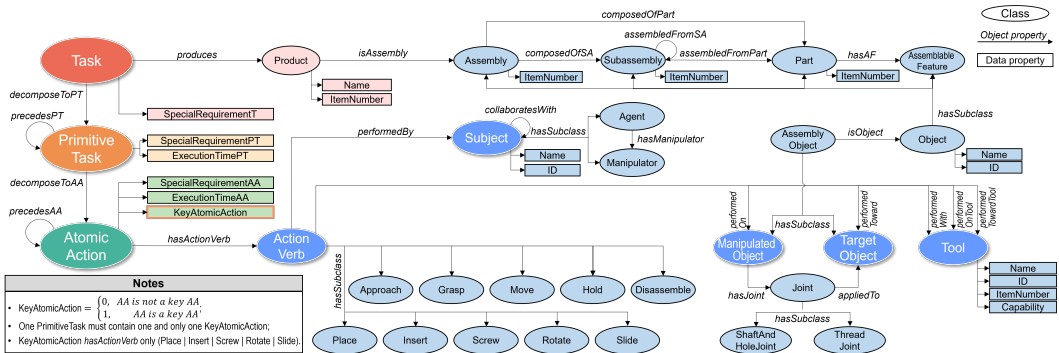

Figure 9: Human-robot shared assembly taxonomy (HR-SAT) schema. We tailored the original taxonomy by removing information that cannot be annotated from videos and incorporating a *Disassembly* action verb to describe human error-and-correction process.

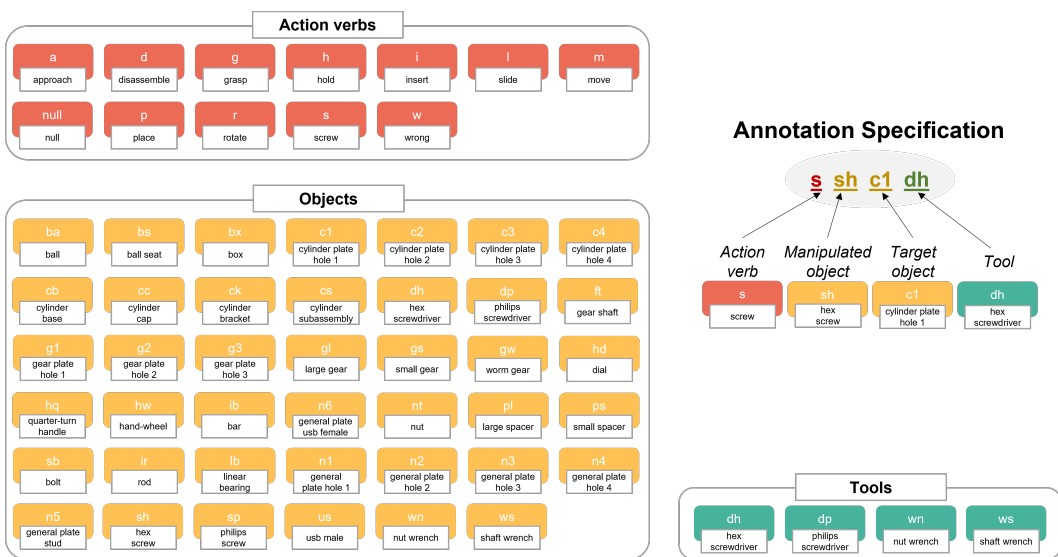

Figure 10: The annotation specification and the list of abbreviated action verbs, objects, and tools annotated in HA-VID.

## 3.1 Action Recognition

We use the MMSkeleton[5] toolbox to benchmark ST-GCN [7]; the MMAction2[6] toolbox to benchmark ST-GCN++ [8], I3D [9], TimeSformer [10], MVITv2 [11], and UniFormerV2 [12]; and the original codes to benchmark TSM [13]. For ST-GCN, we use the joints extracted by the Azure Kinect Body Tracking SDK[7]. However, we only use the upper 26 skeleton joints from the total 32 extracted joints. This excludes the lower limb joints (left and right knee, ankle and foot joints) that are not visible due to the participant standing behind the workbench. Action clips which consisted of frames where the skeleton could not be extracted, were excluded from reporting the performance. For TSM, TimeSformer, I3D (rgb), MVITv2, and UniFormerV2, the RGB frames of each clip were used as input. For I3D (flow), we extracted TV-L1 optical flow frames from each clip as input. To compare model performance on different views (side, front, and top), hands (left and right hands) and annotation levels (primitive task and atomic action), we conducted a combinational benchmark, which means we benchmark each model on 12 sub-datasets (see Figure 15). We report the Top-1 and Top-5 accuracy on these sub-datasets in Table 3.

---

[5]https://github.com/open-mmlab/mmskeleton

[6]https://github.com/open-mmlab/mmaction2

[7]https://learn.microsoft.com/en-us/azure/kinect-dk/body-sdk-download.

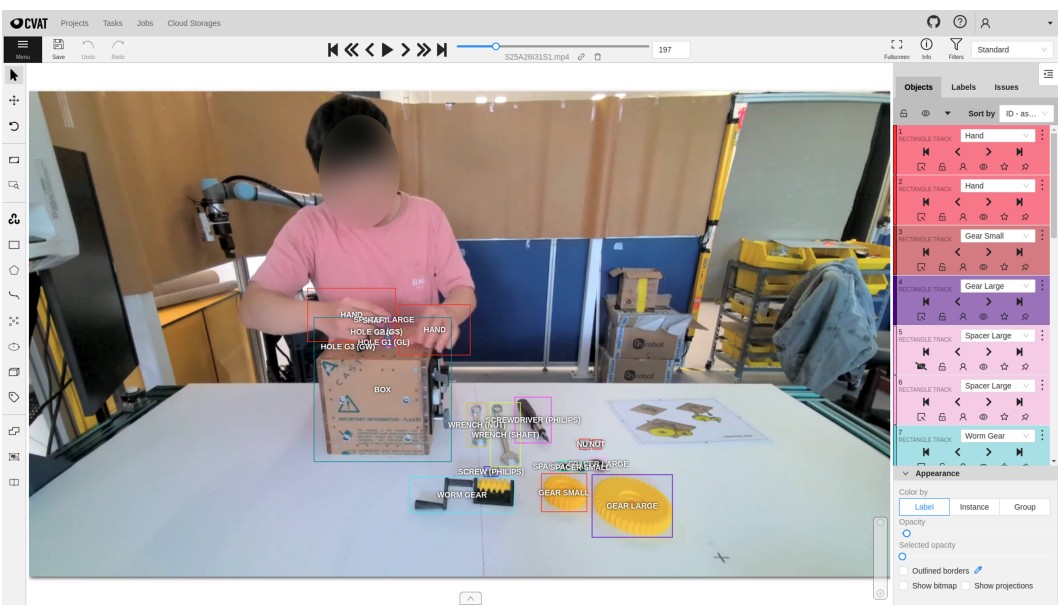

Figure 11: CVAT interface for annotating the subjects (two hands), objects (manipulated object, target object), and tools.

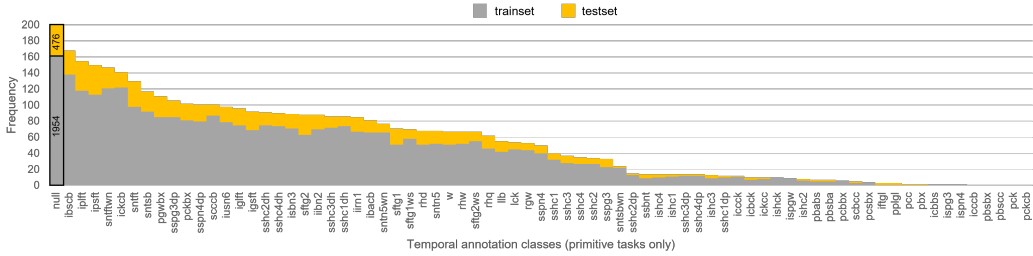

Figure 12: Trainset and testset distribution of the 75 primitive tasks classes. Additionally, to show the distribution better, the frequency axis bound has been reduced, which cuts off the column for the *null* class. Instead, we have manually overwritten the *null* class column with the trainset and testset frequency.

**TSM**: Following the original paper's suggestions, we use the SGD optimizer with a dropout of 0.5. The learning rate was initialized as 0.0025 and decayed by a factor of 10 after epochs 20 and 40. 8 frames were uniformly sampled from each clip. The TSM was pretrained on ImageNet [14], and we finetuned it on our 12 sub-datasets. As the slowest convergence of the 12 sub-datasets was observed around 40 epochs, we set the total training epochs to be 50 with a batch size of 16.

**TimeSformer**: Following the default parameters from MMAction2, we use the SGD optimizer. The learning rate was initialized as 0.005 and decayed by a factor of 10 after epochs 5 and 10. 8 frames were uniformly sampled from each clip. The TimeSformer was pretrained on ImageNet-21K [14], and we finetuned it on our 12 sub-datasets. As the slowest convergence of the 12 sub-datasets was observed around 90 epochs, we set the total training epochs to be 100 with a batch size of 8.

**I3D (rgb) and (flow)**: Following the default parameters from MMAction2, we use the SGD optimizer with a dropout of 0.5. The learning rate was initialized as 0.01 and decayed by a factor of 10 after epochs 40 and 80. 32 frames were uniformly sampled from each clip. I3D takes ResNet50 pretrained on ImageNet-1K [14] as the backbone, and we finetuned it on our 12 sub-datasets. As the slowest convergence of the 12 sub-datasets was observed around 90 epochs, we set the total training epochs to be 100 with a batch size of 4.

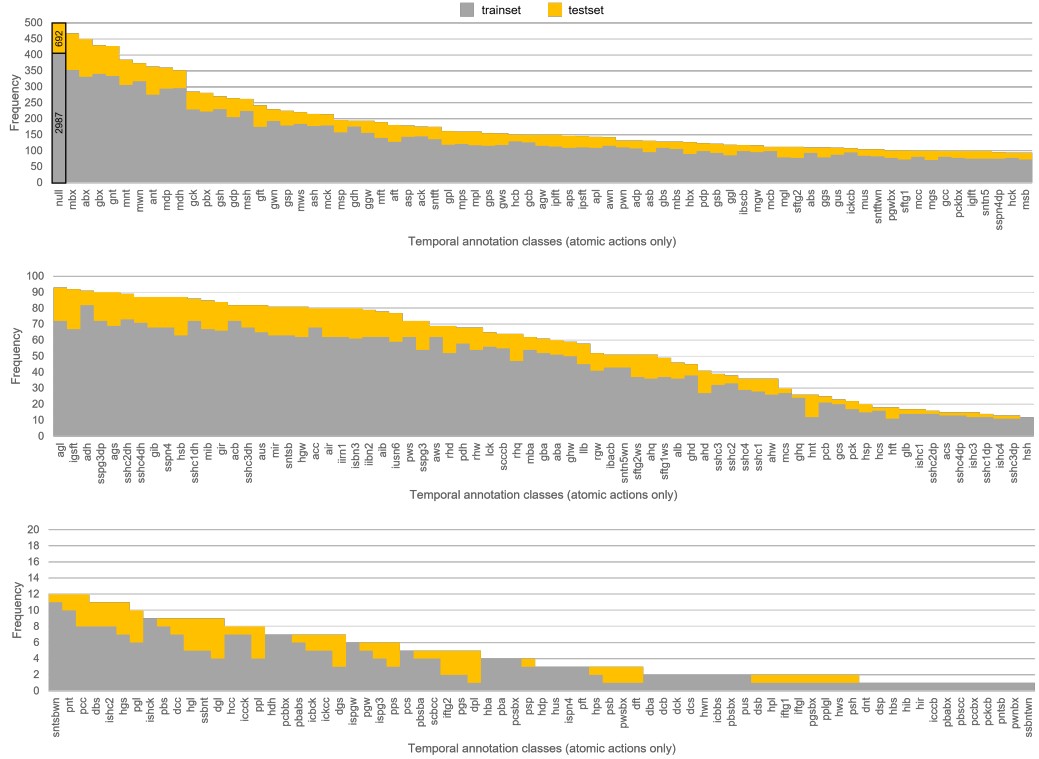

Figure 13: Trainset and testset distribution of the 219 atomic action classes. To show all classes, the diagram is split into three rows. Additionally, to show the distribution better, the frequency axis bound has been reduced, which cuts off the column for the *null* class. Instead, we have manually overwritten the *null* class column with the trainset and testset frequency.

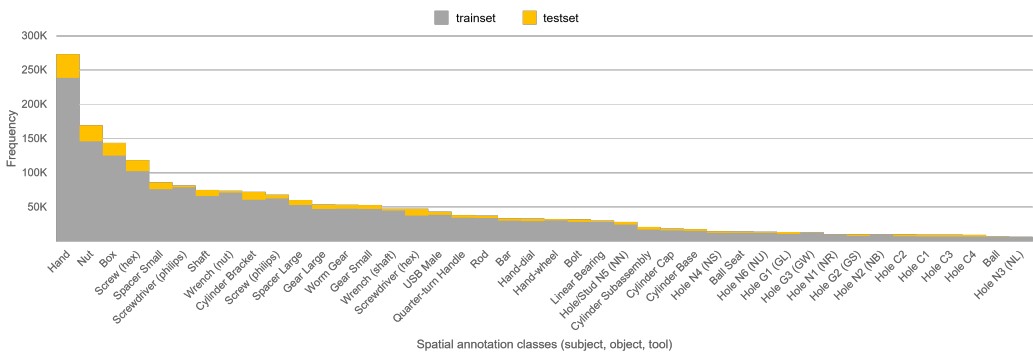

Figure 14: Trainset and testset distribution of the 42 spatial annotation classes. This includes subject, object, and tool.

**MVITv2**: Following the default parameters from MMAction2, we use the AdamW optimizer. Epochs 1 to 20 have a linear learning rate scheduler with an initial learning rate of 0.00015 and start factor of 0.1667. The subsequent epochs use a cosine annealing learning rate with the minimum learning rate ratio of 0.1667. 16 frames were uniformly sampled from each clip. The MVITv2 was pre-trained on Kinetics-400 [15] via MaskFeat [16], and we finetuned it on our 12 sub-datasets. As the slowest convergence of the 12 sub-datasets was observed around 90 epochs, we set the total training epochs to be 100 with a batch size of 4.

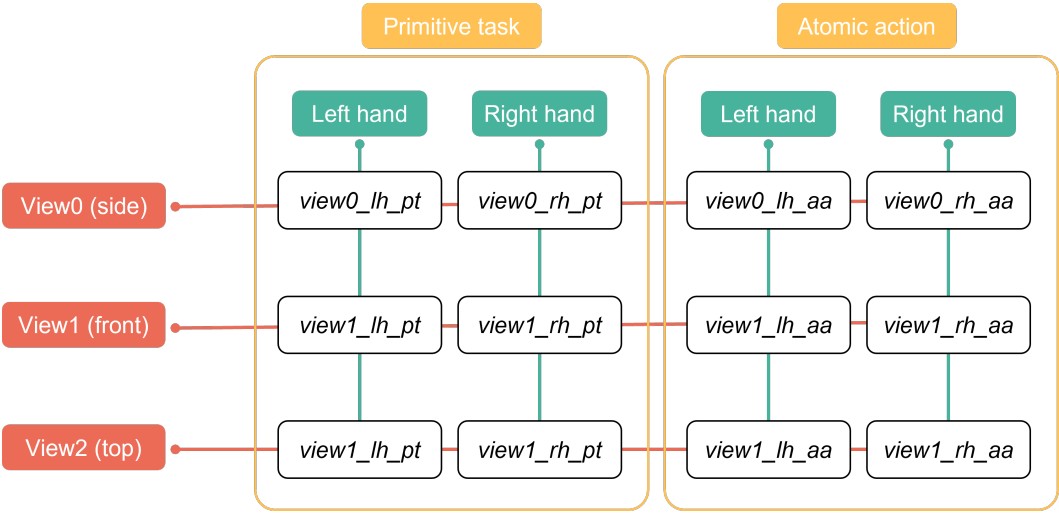

Figure 15: We split the dataset into 12 sub-datasets with three views (side, front, and top), two hands (left and right hands), and two annotation levels (primitive task and atomic action).

**UniFormerV2**: Following the default parameters from MMAction2, we use the AdamW optimizer. Epochs 1 to 5 have a linear learning rate with an initial learning rate of 0.00002 and start factor of 0.05. The subsequent epochs use a cosine annealing learning rate with the minimum learning rate ratio of 0.05. 8 frames were uniformly sampled from each clip. The Uniformerv2 was pre-trained on Kinetics-710 [12], a combined Kinetics-400/600/700 [15] benchmark. As the slowest convergence of the 12 sub-datasets was observed around 20 epochs, we set the total training epochs to be 24 with a batch size of 8.

**ST-GCN**: Following the default parameters from MMSkeleton, we use the SGD optimizer with a dropout of 0.5. The learning rate was initialized as 0.1 and decayed by a factor of 10 after epochs 10 and 50. We sampled all frames as the input. The ST-GCN was pretrained on NTU [17], and we finetuned it on our 12 sub-datasets. As the slowest convergence of the 12 sub-datasets was observed around 70 epochs, we set the total training epochs to be 80 with a batch size of 16.

**ST-GCN++**: Following the default parameters from MMAction2, we use the SGD optimizer. We use a cosine annealing learning rate with the learning rate initialized as 0.0125 and minimum learning rate as 0.100 frames were uniformly samples from each clip. As the slowest convergence of the 12 sub-datasets was observed around 60 epochs, we set the total training epochs to be 80 with a batch size of 16.

The benchmarking results of action recognition are shown in Table 3. We use a single RTX 3090 GPU to train each model, and Table 4 shows the average training time of each model for each sub-dataset.

## 3.2 Action Segmentation

We benchmark three action segmentation algorithms: MS-TCN [18], DTGRM [19], BCN [20], and C2F-TCN [21], and report the frame-wise accuracy (Acc), segmental edit distance (Edit) and segmental F1 score at overlapping thresholds 10% in Table 5. Before benchmarking, we extract I3D features for each frame as the input of the action segmentation algorithms. We use the Pytorch version of the I3D implementation[8] and the pretrained model on ImageNet [14] and Kinetics [15]. For action segmentation, we also conducted a combinational benchmark.

**MS-TCN**: We follow the model settings provided by [18]. More specifically, we use the Adam optimizer with a fixed learning rate of 0.0005, dropout of 0.5 and sampling rate of 1 (taking all frames into the network). As the slowest convergence of the 12 sub-datasets was observed around 800 epochs, we set the total training epochs to be 1000 with a batch size of 10.

---

[8]https://github.com/piergiaj/pytorch-i3d

Table 3: Baselines of action recognition.

| Method | View | Primitive Task | | | | Atomic Action | | | |
| | | Left-Hand | | Right-Hand | | Left-Hand | | Right-Hand | |
| | | Top-1 | Top-5 | Top-1 | Top-5 | Top-1 | Top-5 | Top-1 | Top-5 |
|---|---|---|---|---|---|---|---|---|---|
| TSM | Side | 57.5 | 88.2 | 56.8 | 89.7 | 38.4 | 67.8 | 37.0 | 67.5 |
| | Front | 61.5 | 89.3 | 57.1 | 85.1 | 38.9 | 69.8 | 34.3 | 64.6 |
| | Top | 64.2 | 88.1 | 62.0 | 88.9 | 41.6 | 70.8 | 39.8 | 69.7 |
| TimeSformer | Side | 53.8 | 85.8 | 50.6 | 85.7 | 36.8 | 69.7 | 31.8 | 64.7 |
| | Front | 50.8 | 84.4 | 48.9 | 80.5 | 36.8 | 68.0 | 32.8 | 62.9 |
| | Top | 51.7 | 86.0 | 55.9 | 87.0 | 39.1 | 68.7 | 39.3 | 70.8 |
| I3D (flow) | Side | 38.6 | 50.6 | 37.0 | 44.9 | 23.8 | 46.8 | 23.8 | 45.3 |
| | Front | 39.1 | 54.7 | 37.0 | 45.1 | 23.7 | 48.1 | 23.5 | 46.5 |
| | Top | 39.4 | 57.9 | 37.3 | 48.7 | 22.6 | 45.3 | 23.9 | 45.9 |
| I3D (rgb) | Side | 54.9 | 82.5 | 51.8 | 83.7 | 38.2 | 72.0 | 34.0 | 66.8 |
| | Front | 52.8 | 83.6 | 51.6 | 82.9 | 41.6 | 73.5 | 35.6 | 66.0 |
| | Top | 54.4 | 85.0 | 57.6 | 84.0 | 41.3 | 70.3 | 41.2 | 71.3 |
| I3D (both) | Side | 32.2 | 45.7 | 51.1 | 85.2 | 40.8 | 75.6 | 37.6 | 71.4 |
| | Front | 53.2 | 83.6 | 49.7 | 84.4 | 44.0 | 75.9 | 39.6 | 71.3 |
| | Top | 57.7 | 85.0 | 57.8 | 85.6 | 44.1 | 73.5 | 44.4 | 75.9 |
| MVITv2 | Side | 58.5 | 85.2 | 57.8 | 85.2 | **48.5** | 76.5 | 41.8 | 70.8 |
| | Front | **63.1** | 86.6 | 55.9 | 81.6 | **48.3** | 76.4 | 41.9 | 70.1 |
| | Top | 62.9 | 87.1 | 62.5 | 85.4 | 48.3 | 76.5 | **44.9** | 72.8 |
| UniFormerV2 | Side | 58.4 | **88.6** | **58.7** | **91.1** | 46.3 | **79.8** | **48.8** | **80.1** |
| | Front | 62.5 | **90.4** | **59.1** | **89.1** | 48.3 | **79.6** | 43.4 | **76.5** |
| | Top | **66.4** | **90.1** | **66.4** | **89.5** | **52.2** | **83.5** | 41.5 | **76.1** |
| ST-GCN | Side | 40.7 | 61.5 | 41.4 | 61.3 | 22.2 | 46.0 | 21.5 | 44.4 |
| | Front | 41.9 | 65.7 | 39.3 | 57.7 | 21.9 | 46.6 | 19.9 | 40.5 |
| | Top | 35.8 | 53.4 | 35.4 | 46.7 | 16.8 | 40.7 | 17.8 | 36.9 |
| ST-GCN++ | Side | 40.6 | 59.4 | 19.6 | 43.6 | 39.3 | 60.4 | 17.6 | 38.8 |
| | Front | 42.4 | 62.2 | 19.6 | 39.3 | 38.3 | 55.4 | 17 | 34.8 |
| | Top | 33.3 | 52.5 | 17.9 | 40.9 | 34.9 | 54.2 | 15.5 | 34.6 |

Table 4: Training efficiency of ST-GCN, TSM, TimeSformer, I3D, and MVITv2.

| Dataset | | | Average training time per epoch (min) | | | | | | | |
| View | Hand | Task level | ST-GCN | ST-GCN++ | TSM | TimeSformer | I3D (flow) | I3D (rgb) | MVITv2 | UniFormerV2 |
|---|---|---|---|---|---|---|---|---|---|---|
| Side | Left hand | Primitive task | 1.65 | 2.25 | 1.3 | 6.12 | 3.3 | 5.83 | 11.12 | 3.95 |
| | | Atomic action | 5.55 | 7.12 | 2.6 | 14.42 | 10.82 | 10.02 | 24.9 | 12.83 |
| | Right hand | Primitive task | 1.73 | 2.02 | 1.4 | 4.2 | 4.22 | 5.72 | 6.95 | 3.79 |
| | | Atomic action | 5.38 | 6.14 | 4.48 | 12.85 | 9.12 | 11.73 | 23.55 | 11.55 |
| Front | Left hand | Primitive task | 1.73 | 2.13 | 1.33 | 3.93 | 4.15 | 5.88 | 11.15 | 3.97 |
| | | Atomic action | 5.72 | 6.86 | 4.5 | 21.4 | 9.63 | 12.23 | 25.37 | 12.67 |
| | Right hand | Primitive task | 1.82 | 2.05 | 1.22 | 4.22 | 2.48 | 4.68 | 6.98 | 3.74 |
| | | Atomic action | 5.65 | 6.14 | 4.27 | 12.82 | 7.02 | 11.18 | 26.58 | 11.45 |
| Top | Left hand | Primitive task | 0.71 | 1.24 | 1.38 | 4.08 | 5.25 | 5.55 | 11.5 | 3.87 |
| | | Atomic action | 3.01 | 5.14 | 4.75 | 14.3 | 10.05 | 11.57 | 24.05 | 12.54 |
| | Right hand | Primitive task | 0.65 | 1.12 | 1.4 | 4.17 | 4.47 | 2.8 | 8.33 | 3.65 |
| | | Atomic action | 2.43 | 5.21 | 4.57 | 12.8 | 7.07 | 10.93 | 24.03 | 11.20 |

**DTGRM**: We follow the model settings provided by [19]. More specifically, we use the Adam optimizer with a fixed learning rate of 0.0005, dropout of 0.5 and sampling rate of 1. As the slowest convergence of the 12 sub-datasets was observed around 800 epochs, we set the total training epochs to be 1000 with a batch size of 16.

**BCN**: We follow the model settings provided by [20]. More specifically, we use the Adam optimizer with the learning rate of 0.001 for the first 30 epochs and 0.0001 for the rest epochs, dropout of 0.5 and sampling rate of 1. As the slowest convergence of the 12 sub-datasets was observed around 200 epochs, we set the total training epochs to be 300 with a batch size of 1.

**C2F-TCN**: We follow the model settings provided by [21]. More specifically, we use the Adam optimizer with the learning rate of 0.00005 and sampling rate of 1. As the slowest convergence of the 12 sub-datasets was observed around 1000 epochs, we set the total training epochs to be 1500 with a batch size of 100.

The benchmarking results of action segmentation are shown in Table 5. We use a single RTX 3090 GPU to train each model, and Table 6 shows the average training time of each model for each sub-dataset.

Table 5: Baselines of Action Segmentation.

| Method | View | Primitive task | | | | | | Atomic Action | | | | | |
| | | Left hand | | | Right hand | | | Left hand | | | Right hand | | |
| | | F1 | Edit | Acc | F1 | Edit | Acc | F1 | Edit | Acc | F1 | Edit | Acc |
|---|---|---|---|---|---|---|---|---|---|---|---|---|---|
| | Side | 37.6 | 37.4 | 41.2 | 31.1 | 32.5 | 37.4 | **35.3** | **32.1** | 40.9 | 29.2 | **31.0** | 32.6 |
| MS-TCN | Front | 35.2 | 36.3 | 38.8 | 36.7 | 36.2 | 39.3 | **34.1** | 31.2 | **41.1** | 29.3 | **31.1** | 33.4 |
| | Top | 37.1 | 38.9 | 40.4 | 36.1 | 35.6 | 41.3 | **35.9** | **34.1** | 40.8 | 35.1 | **34.6** | **37.8** |
| | Side | 38.5 | 36.5 | 40.9 | 35.9 | 35.2 | 37.6 | 33.7 | 30.7 | 39.3 | 27.8 | 28.2 | 30.3 |
| DTGRM | Front | 38.5 | 37.2 | 39.0 | 38.8 | **39.6** | 40.5 | 34.0 | **33.6** | 39.7 | 27.6 | 27.8 | 31.5 |
| | Top | 40.4 | 38.8 | 40.8 | 38.7 | 37.0 | 41.2 | 35.1 | 33.6 | 40.5 | 34.0 | 31.9 | 37.6 |
| | Side | **43.1** | **40.4** | **43.7** | **38.6** | **36.3** | **42.4** | 21.3 | 18.0 | 39.5 | 20.5 | 18.9 | **34.1** |
| BCN | Front | **44.4** | **43.1** | **44.4** | **41.3** | 37.0 | **44.0** | 17.2 | 14.4 | 39.5 | 22.9 | 20.7 | **34.3** |
| | Top | **43.5** | **40.7** | **44.3** | **44.0** | **40.7** | 43.7 | 16.8 | 15.3 | 40.1 | 23.4 | 20.6 | 35.5 |
| | Side | 18.0 | 20.7 | 38.2 | 21.7 | 20.9 | 38.8 | 19.9 | 18.2 | 37.9 | 15.8 | 15.7 | 30.3 |
| C2F-TCN | Front | 23.3 | 21.4 | 39.1 | 20.5 | 22.2 | 38.8 | 20.1 | 19.8 | 37.4 | 17.9 | 17.5 | 31.4 |
| | Top | 26.6 | 24.0 | 41.1 | 25.4 | 22.7 | 39.5 | 21.5 | 19.7 | 37.7 | 18.6 | 18.9 | 33.6 |

Table 6: Training efficiency of MS-TCN, DTGRM and BCN.

| Dataset | | | Average training time per epoch (sec) | | | |
| View | Hand | Task level | MS-TCN | DTGRM | BCN | C2F-TCN |
|---|---|---|---|---|---|---|
| Side | Left hand | Primitive task | 8.24 | 18.66 | 16.35 | 13.47 |
| | | Atomic action | 8.37 | 19.42 | 16.50 | 13.67 |
| | Right hand | Primitive task | 8.86 | 20.01 | 16.26 | 13.64 |
| | | Atomic action | 8.66 | 20.41 | 16.51 | 13.85 |
| Front | Left hand | Primitive task | 8.04 | 19.44 | 16.31 | 13.76 |
| | | Atomic action | 8.01 | 19.82 | 16.38 | 13.68 |
| | Right hand | Primitive task | 8.31 | 20.05 | 16.24 | 13.76 |
| | | Atomic action | 8.45 | 19.12 | 16.56 | 13.83 |
| Top | Left hand | Primitive task | 7.81 | 19.44 | 16.39 | 13.72 |
| | | Atomic action | 7.97 | 19.44 | 16.42 | 14.09 |
| | Right hand | Primitive task | 8.23 | 18.70 | 16.31 | 13.66 |
| | | Atomic action | 8.30 | 19.27 | 16.51 | 13.94 |

## 3.3 Object Detection

We benchmark three object detection algorithms: Faster-RCNN [22], YOLOv5 [23] and DINO [24] with different backbone networks. The results have been reported in the main paper. Therefore, we only discuss the implementation details here. We train Faster-RCNN and DINO using the implementation provided by the MMDetection [25] and train YOLOv5 using the implementation provided by the MMYOLO[9].

**Faster-RCNN**: We train Faster-RCNN with three backbone networks: ResNet50, ResNet101, and ResNext101. All the networks have been pretrained on the coco_2017_train dataset [26] and finetuned on our dataset. Following the default setting provided by MMDetection, we use the SGD optimizer with a momentum of 0.9 and weight decay of 0.0001. The learning rate was initialized as 0.02 and decayed by a factor of 10 at epochs 8 and 11. As the slowest convergence of the three models was observed around 14 epochs, we set the total training epochs to be 20. We set the batch size as 4, 1, and 5, respectively, for ResNet50, ResNet101, and ResNext101.

**YOLOv5**: We train YOLOv5-small and YOLOv5-large using MMDetection. These two models have been pretrained on the coco_2017_train dataset, and finetuned on our dataset. Following the default setting provided by MMDetection, we use the SGD optimizer with a momentum of 0.937, weight decay of 0.0005 for both models. The linear learning rate with base learning rate of 0.0025 and factor of 0.01 was applied to YOLOv5-small. The linear learning rate with base learning rate of 0.0025 and factor of 0.1 was applied to YOLOv5-large. We set the total training epochs to be 100 epochs with a batch size of 32 and 50 epochs with a batch size of 10, respectively, for YOLOv5-small and YOLOv5-large to ensure convergence.

[9]https://github.com/open-mmlab/mmyolo

**DINO**: We benchmark the DINO model with the Swin-large network as the backbone. The model has been pretrained on the coco_2017_train dataset, and finetuned on our dataset. Following the default setting provided by MMDetection, we use the AdamW optimizer with a learning rate of 0.0001 and weight decay of 0.0001. As the convergence was observed around 6 epochs, we set the total training epochs to be 10 with a batch size of 1.

We use single RTX 3090 GPU to train each model, and Table 7 shows the average training time of each model.

Table 7: Training efficiency of Faster-RCNN, YOLOv5 and DINO.

| Method | | Average training time per epoch (min) |
| --- | --- | --- |
| | ResNet50 | 446.9 |
| Faster-RCNN | ResNet101 | 197.0 |
| | ResNext101 | 668.8 |
| YOLOv5-s | DarkNet | 39.5 |
| YOLOv5-l | DarkNet | 94.2 |
| DINO | Swin-L | 1592.3 |

## 3.4 Multi-Object Tracking

In this paper, we focus on tracking-by-detection methods because, normally, tracking-by-detection methods perform better than joint-detection-association methods [27]. Since we already benchmarked the object detection methods, we only need to test the SOTA trackers. We benchmark SORT [28] and ByteTrack [29] trackers on the detection results of DINO and ground truth annotations, respectively. The results have been reported in the main paper. Since the trackers are not neural networks, we do not need to train them and explain the implementation details. We always use the default parameters of the algorithm. For more details, please refer to the papers [28, 29] and their GitHub repositories.

## 3.5 Discussion

In this section, we further discuss the results from the above experiments, examine the action anticipation task, investigate class imbalance problem, and analyze a prevalent problem of video understanding – occlusion.

### 3.5.1 General Discussion

**Action recognition**: We found the Top-1 accuracy of primitive task recognition is 15.6% higher on average than atomic action recognition, and the atomic action recognition performance of the left hand is 2.4% higher on average than the right hand. One possible reason behind these two observations can be occlusion since (1) primitive task recognition is less influenced by occlusion because it can rely on the key motion or relevant object recognition; and (2) the left hand is less occluded because the side-view camera is mounted on the left-side of the participant. We also found the accuracy of skeleton-based accuracy is significantly lower than the other action recognition benchmarks. One possible reason could be that skeleton-based action recognition only considers human motion and does not take visual object information as input. Another possible reason could be the uncertain quality of the skeleton data, since the ground truth skeleton data is difficult to obtain.

**Action segmentation**: We found (1) the frame-wise accuracy (Acc) of atomic action segmentation is 4% lower on average than primitive task segmentation, as atomic actions have higher diversity and current methods face under-segmentation issues (refer to the main paper); and (2) on the atomic action level, the Acc of the left hand is 6% higher on average than the right hand, where one possible reason could be that the left hand is less occluded.

**Object detection**: From Table 4 of the main paper, we found that (1) the large-scale end-to-end Transformer based model (DINO) performs the best, and the traditional two-stage method (Faster-RCNN) has better performance on small objects but worse performance on large objects than the one-stage method (YOLOv5), which is consistent with the conclusion of [30]; (2) current methods still face great challenges in small object detection, as the best model only has 27.4% average precision on small object detection; and (3) recognizing objects with same/similar appearances but different sizes is challenging (see Figure 16, e.g., Bar and Rod, Hole C1-C4, and two Wrenches).

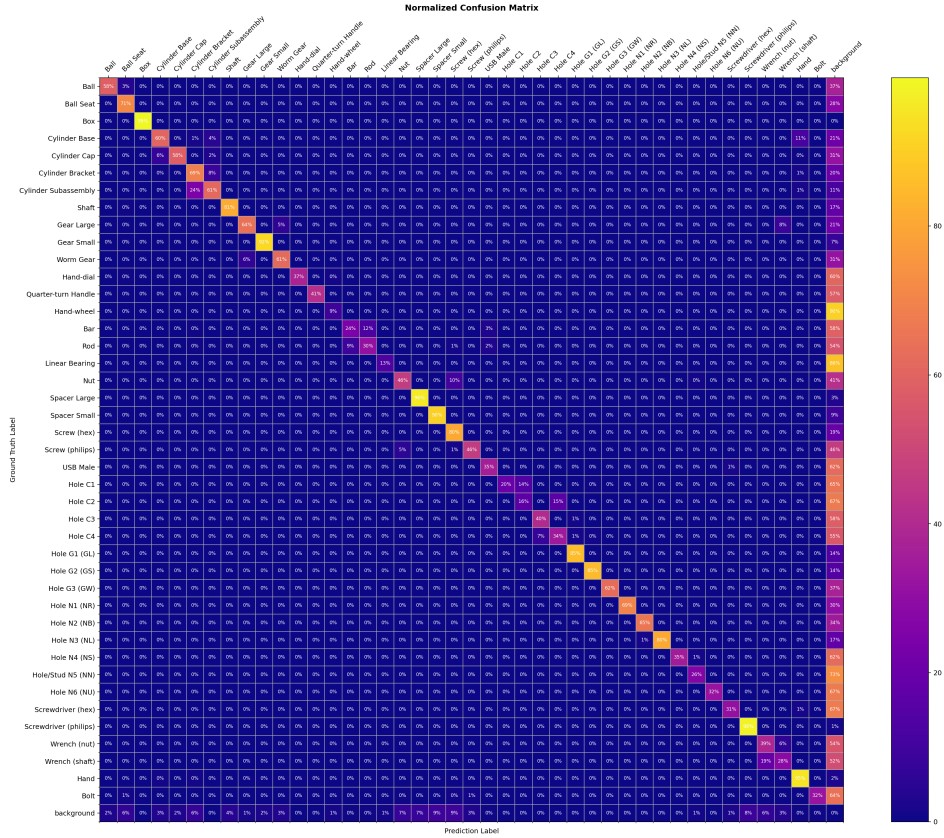

Figure 16: Confusion matrix of object detection results from DINO.

**Multi-object detection**: From Table 5 of the main paper, we found that (1) object detection performance is the decisive factor in tracking performance; (2) with perfect detection results, even the simple tracker (SORT) can achieve good tracking results, as SORT has 94.5% multi-object tracking accuracy on the ground truth object bounding boxes; and (3) ByteTrack can track blurred and occluded objects better (comparing b1-2, c1-2, and f1-2 in Figure 17) due to taking low-confidence detection results into association, but it generates more ID switches (IDS) (seeing a2-f2 in Figure 17) due to the preference of creating new tracklets.

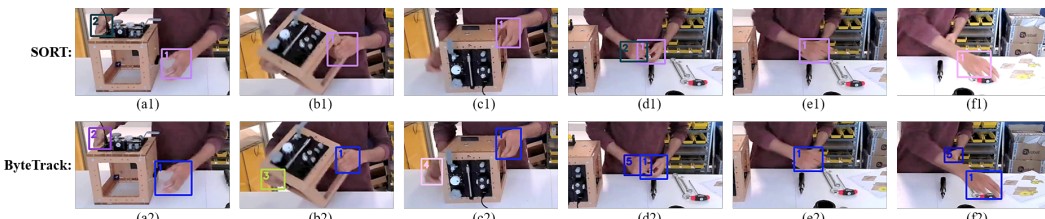

Figure 17: Visualizing the tracking results of SORT and ByteTrack (taking hand as an example).

### 3.5.2 Action Anticipation

Recognizing the indispensability of assembly video reasoning in comprehending application-oriented knowledge, we conducted a preliminary exploration of a typical video reasoning task, action anticipation, on HA-ViD. We benchmark one long-term action anticipation algorithm – FUTR [31]. The problem setting is that for a video with $T$ frames, the first $\alpha T$ frames are observed and a sequence of actions for the next $\beta T$ frames is anticipated. We use the pre-extracted I3D features (same as action segmentation) as input visual features. We follow the model settings provided by [31]. In

training, we set the observation rate $\alpha \in \{0.2, 0.3, 0.5\}$ and fix the prediction rate $\beta$ to 0.5. We use AdamW optimizer with a learning rate of 0.0001, dropout of 0.5 and sampling rate of 1. We employ a cosine annealing warm-up scheduler with warm-up stages of 10 epochs and base learning rate of 0.0003. As the slowest convergence of the 12 sub-datasets was observed around 150 epochs, we set the total training epochs to be 200 with a batch size of 16. In inference, we set the observation rate $\alpha \in \{0.2, 0.3\}$ and prediction rate $\beta \in \{0.1, 0.2, 0.3, 0.5\}$ and measure mean over classes (MoC) accuracy following the long-term action anticipation framework protocol [32]. To compare the performance difference between anticipating action (action verb + manipulated object + target object + tool), anticipating action verb only and anticipating manipulated object only, we conduct three sets of experiments and report the results in Table 8. We found that the main factor affecting action anticipation performance is the task complexity, as there are 75 action anticipation, 7 action verb anticipation, and 25 manipulated object anticipation classes on the primitive task level and 219 action anticipation, 11 action verb anticipation, and 29 manipulated object anticipation classes on the atomic action level.

Table 8: Comparison between action anticipation, action verb anticipation and manipulated object anticipation.

| | Task level | Hand | View | $\beta$ ($\alpha = 0.2$) | | | | $\beta$ ($\alpha = 0.3$) | | | |
|---|---|---|---|---|---|---|---|---|---|---|---|
| | | | | 0.1 | 0.2 | 0.3 | 0.5 | 0.1 | 0.2 | 0.3 | 0.5 |
| Action anticipation | Primitive task | Left-hand | Side | 7.9 | 5.1 | 4.5 | 3.3 | 13.5 | 10.1 | 6.7 | 5.2 |
| | | | Front | 5.9 | 3.3 | 2.7 | 4.6 | 12 | 8.7 | 6.5 | 5.8 |
| | | | Top | 7.5 | 7.5 | 6.4 | 6.2 | 15.3 | 10.9 | 8.5 | 8 |
| | | Right-hand | Side | 6.7 | 4.7 | 3.6 | 3.4 | 12.2 | 10.2 | 6.7 | 6.6 |
| | | | Front | 5.9 | 7.1 | 5 | 4.8 | 10.4 | 5.8 | 3.1 | 2.9 |
| | | | Top | 10.4 | 8.5 | 6 | 4.3 | 10.3 | 9.3 | 7.7 | 5.1 |
| | Atomic action | Left-hand | Side | 3.4 | 2.2 | 2 | 1.8 | 2.9 | 3.2 | 2.9 | 1.9 |
| | | | Front | 2.6 | 1.9 | 1.9 | 1.6 | 5 | 3.3 | 2.8 | 2.3 |
| | | | Top | 2.5 | 1.7 | 1.8 | 1.5 | 3.9 | 2.5 | 2.2 | 2.3 |
| | | Right-hand | Side | 1.4 | 1.8 | 1.7 | 1.3 | 4.5 | 2.9 | 2.4 | 1.6 |
| | | | Front | 1.7 | 1.6 | 1.3 | 1.2 | 6.1 | 2.1 | 1.5 | 1.2 |
| | | | Top | 1.7 | 1.6 | 1.3 | 1.2 | 6.1 | 2.1 | 1.5 | 1.2 |
| Action verb anticipation | Primitive task | Left-hand | Side | 12.5 | 14.5 | 19.9 | 18.8 | 25.7 | 16.6 | 16.1 | 15.3 |
| | | | Front | 18.5 | 19.5 | 19 | 17.6 | 30.6 | 18.1 | 17.4 | 16.1 |
| | | | Top | 17.9 | 17.5 | 17.6 | 17.2 | 28.5 | 20.6 | 19.5 | 17.9 |
| | | Right-hand | Side | 24.3 | 21.7 | 16.9 | 16.6 | 35.7 | 19 | 16.8 | 16.2 |
| | | | Front | 33.8 | 31.3 | 17.9 | 17.9 | 38.1 | 18.2 | 14.1 | 14.4 |
| | | | Top | 38.2 | 36.5 | 17.8 | 17.7 | 38.4 | 19 | 15.9 | 16.1 |
| | Atomic action | Left-hand | Side | 12 | 10.3 | 9.5 | 9.7 | 14.2 | 10.8 | 9.4 | 9.5 |
| | | | Front | 13.5 | 12.9 | 12 | 10.8 | 15.2 | 13.5 | 11.1 | 9.6 |
| | | | Top | 12.5 | 9.6 | 8.8 | 9.7 | 17.6 | 14.3 | 12.8 | 11 |
| | | Right-hand | Side | 14.9 | 11.1 | 11.2 | 10.5 | 15.7 | 13.2 | 11.6 | 10 |
| | | | Front | 11.7 | 9.7 | 9.7 | 9.7 | 14.5 | 12.9 | 13 | 10.7 |
| | | | Top | 23.1 | 18.4 | 12.9 | 12 | 15.2 | 13.1 | 12 | 10.4 |
| Manipulated object anticipation | Primitive task | Left-hand | Side | 6.8 | 6.4 | 6.5 | 7.7 | 16.8 | 12.9 | 12.2 | 7.7 |
| | | | Front | 10.7 | 6.1 | 7 | 8.4 | 16 | 14.4 | 14.5 | 10.6 |
| | | | Top | 13.5 | 10.4 | 9.4 | 11.6 | 12.5 | 9.7 | 10 | 10.4 |
| | | Right-hand | Side | 15.9 | 11.6 | 9.4 | 8.6 | 19.7 | 13.1 | 8 | 6.6 |
| | | | Front | 13.8 | 10.3 | 11.8 | 10.7 | 20.7 | 13.3 | 11.7 | 11.7 |
| | | | Top | 20.3 | 15 | 10.6 | 10.8 | 16.5 | 14 | 8 | 7.6 |
| | Atomic action | Left-hand | Side | 13.1 | 11.8 | 10.9 | 10.7 | 11.6 | 8.9 | 8.2 | 8.9 |
| | | | Front | 10.6 | 9.5 | 10.3 | 9.7 | 15.8 | 13.2 | 12.7 | 9.6 |
| | | | Top | 10.3 | 10 | 10 | 10.2 | 18.2 | 17 | 15 | 11.6 |
| | | Right-hand | Side | 4.3 | 4.7 | 4.9 | 3.9 | 11.1 | 7.6 | 7 | 6.4 |
| | | | Front | 6.9 | 9.6 | 7.3 | 6.5 | 13.4 | 9.1 | 7.9 | 7.3 |
| | | | Top | 9 | 9.1 | 8.7 | 7.5 | 10.8 | 8.5 | 8.3 | 5.8 |

### 3.5.3   Class Imbalance

The statistics of HA-ViD reveal a distinct class imbalance problem, a challenge often encountered in real-world. Taking a step to mitigate this problem, we utilize an over-sampling strategy on our dataset, randomly over-sampling the minority classes (sample size less than a threshold) to reach the threshold. Then, we test the performance of UniFormerV2 on the over-sampled dataset. The implementation details of UniFormerV2 are the same as described in Section 3.1. After a preliminary experiment on different thresholds, we set the threshold set to 300. The results are reported in Table 9. Compared with the original dataset, there is a noticeable improvement of Top-1 accuracy in the over-sampled dataset.

Table 9: Comparison between the results of UniFormerV2 on the original dataset and over-sampled dataset.

| Method | View | Primitive Task | | | | Atomic Action | | | |
| | | Left-Hand | | Right-Hand | | Left-Hand | | Right-Hand | |
| | | Top-1 | Top-5 | Top-1 | Top-5 | Top-1 | Top-5 | Top-1 | Top-5 |
|---|---|---|---|---|---|---|---|---|---|
| | Side | 58.4 | 88.6 | 58.7 | **91.1** | 46.3 | 79.8 | 48.8 | **80.1** |
| UniFormerV2 | Front | 62.5 | **90.4** | 59.1 | **89.1** | 48.3 | 79.6 | 43.4 | 76.5 |
| | Top | 66.4 | **90.1** | 66.4 | 89.5 | 52.2 | **83.5** | 41.5 | 76.1 |
| | Side | **66.9** | **90.1** | 65.1 | 89.8 | **51.3** | **80.7** | **50.9** | 79.9 |
| UniFormerV2 (Over-sampling) | Front | **66.9** | 88.5 | 64.3 | 87.1 | **50.3** | 79.8 | 44.7 | 74.8 |
| | Top | **68.1** | 89.1 | 69.2 | **90.3** | **54.5** | 83 | **51** | **79.6** |

### 3.5.4 Occlusion Analysis

From the discussion in Section 3.5.1, we can see occlusion is a prevalent problem of video understanding. Therefore, we further explore the impact of occlusion on video understanding tasks in this Section. Table 10 reports the average results over two hands of action recognition and segmentation on three views and the combined view (Com). We fuse the features from three views before the softmax layer to evaluate the performance of the combined view. The results show the significant benefits of combining three views which offers a viable solution for mitigating occlusion challenges in industrial settings.

Table 10: Performance of action recognition and segmentation on three views and the combined view.

| View | Action Segmentation (BCN) | | | | | | Action Recognition (MVITv2) | | | |
| | Primitive task | | | Atomic action | | | Primitive task | | Atomic action | |
| | F1 | Edit | Acc | F1 | Edit | Acc | Top-1 | Top-5 | Top-1 | Top-5 |
|---|---|---|---|---|---|---|---|---|---|---|
| Side | 40.9 | 38.4 | 43.1 | 20.9 | 18.5 | 36.8 | 58.2 | 85.2 | 45.2 | 73.7 |
| Front | 42.9 | 40.1 | 44.2 | 20.1 | 17.6 | 36.9 | 59.5 | 84.1 | 45.1 | 73.3 |
| Top | 43.8 | 40.7 | 44 | 20.1 | 18.0 | 37.8 | 62.7 | 86.3 | 46.6 | 74.7 |
| Com | **44.6** | **45.9** | **47.2** | **41.7** | **35.9** | **44.5** | **64.0** | **89** | **50.8** | **80.9** |

Figure 17 shows the impact of occlusion on tracking and reidentification via visualizing SORT and ByteTrack tracking results on sampled ground truth object annotations. To quantitatively analyze the occlusion problem, we design two metrics: occlusion duration (OD) and occlusion frequency (OF). Given a video of $n$ frames $v = [f_1, \ldots, f_n]$, the observation of object $k$ is denoted as $O_k = [o_t^k, o_{t+1}^k, \ldots, o_{t+m}^k]$, where $t$ and $t + m$ are the frame numbers that object $k$ first, and last appear, respectively. $o_j^k = \{0, 1\}$, where 0 denotes observed, and 1 denotes unobserved. $OD_k = \frac{1}{m} \sum_{j=t}^{j=t+m} o_j^k$ and $OF_k = \frac{1}{2} \sum_{j=t}^{j=t+m-1} |o_{j+1}^k - o_j^k|$. $OD_k$ and $OF_k$ describe the occluded duration and occluded frequency of object $k$ in a video. We calculate the average OD and OF over every object in our testing dataset and compare the results with the tracking results on ground truth object annotations in Table 11. Table 11 shows a negative correlation between mOD and mOF with MOTA and IDS, which is also consistent with the findings in Figure 17. We envision OD and OF will serve as effective occlusion evaluation tools for developing better object association modules and reidentification modules in MOT.

### 3.6 Licenses of the benchmarked algorithms

The licenses of the benchmarked algorithms are listed in Table 12.

Table 11: Comparison between tracking results and occlusion metrics on three views.

| View | Method | MOTA | IDF1 | IDS | mOD | mOF |
|---|---|---|---|---|---|---|
| Side | SORT | 93.5% | 66.5% | 58.3 | 18.7% | 4.1 |
| | ByteTrack | 98.5% | 68.4% | 124.5 | | |
| Front | SORT | 95.3% | 72.1% | 48.2 | 12.1% | 2.9 |
| | ByteTrack | 98.7% | 67.8% | 118.7 | | |
| Top | SORT | 94.7% | 68.6% | 57.8 | 14.7% | 5.3 |
| | ByteTrack | 98.4% | 66.3% | 121.5 | | |

Table 12: Licenses of the benchmarked algorithms.

| Algorithm | License |
|---|---|
| MMSkeleton | Apache License 2.0 |
| ST-GCN | BSD 2-Clause "Simplified" License |
| MMAction2 | Apache License 2.0 |
| ST-GCN++ | Apache License 2.0 |
| TSM | MIT |
| TimeSFormer | Attribution-NonCommercial 4.0 International |
| I3D | Apache License 2.0 |
| MVITv2 | Apache License 2.0 |
| UniFormerV2 | Apache License 2.0 |
| MS-TCN | MIT |
| DTGRM | MIT |
| BCN | MIT |
| C2F-TCN | MIT |
| FUTR | MIT + Commons Clause License Condition v1.0 |
| MMDetection | Apache License 2.0 |
| Faster-RCNN | MIT |
| DINO | Apache License 2.0 |
| MMYOLO | GNU General Public License v3.0 |
| YOLOv5 | GNU Affero General Public License v3.0 |
| SORT | GNU General Public License v3.0 |
| ByteTrack | MIT |

## 4 Dataset Bias and Societal Impact

Our objective is to construct a dataset that can represent interesting and challenging problems in real-world industrial assembly scenarios. Based on this objective, we developed the Generic Assembly Box that encompasses standard and non-standard parts widely used in industry and requires typical industrial tools to assemble. However, there is still a gap between our dataset and the real-world industrial assembly scenarios. The challenges lie in:

1) the existence of numerous unique assembly actions, countless parts, and tools in the industry;

2) the vast diversity of operating environments in the industry;

3) various agents and multi-agent collaborative assembly scenarios in the industry.

Therefore, additional efforts would be needed to apply the models trained on our dataset to real-world industrial applications. We hope the fine-grained annotations of this dataset can advance the technological breakthrough in comprehensive assembly knowledge understanding from videos. Then, the learned knowledge can benefit various real-world applications, such as robot skill learning, human-robot collaboration, assembly process monitoring, assembly task planning, and quality assurance. We hope this dataset can contribute to technological advancements facilitating the development of smart manufacturing, enhancing production efficiency, and reducing the workload and stress on workers.

## 5 Ethics Approval

HA-ViD was collected with ethics approval from the University of Auckland Human Participants Ethics Committee. The Reference Number is 21602. All participants were sent a Participant Information Sheet and Consent Form[10] prior to the collection session. We confirmed that they had agreed to and signed the Consent form before proceeding with any data collection.

## 6 Data Documentation

We follow the datasheet proposed in [33] for documenting our HA-ViD dataset:

---

[10]The participant consent form is available at: `https://www.dropbox.com/sh/ekjle5bwoylmdcf/AACLd_NqT3p2kxW7zLvvauPta?dl=0`

1. Motivation

(a) For what purpose was the dataset created?

This dataset was created to understand comprehensive assembly knowledge from videos. The previous assembly video datasets fail to (1) represent real-world industrial assembly scenarios, (2) capture natural human behaviors (varying efficiency, alternative routes, pauses and errors) during procedural knowledge acquisition, (3) follow a consistent annotation protocol that aligns with human and robot assembly comprehension.

(b) Who created the dataset, and on behalf of which entity?

This dataset was created by Hao Zheng, Regina Lee and Yuqian Lu. At the time of creation, Hao and Regina were PhD students at the University of Auckland, and Yuqian was a senior lecturer at the University of Auckland.

(c) Who funded the creation of the dataset?

The creation of this dataset was partially funded by The University of Auckland FRDF New Staff Research Fund (No. 3720540).

(d) Any other Comments?

None.

2. Composition

(a) What do the instances that comprise the dataset represent?

For the video dataset, each instance is a video clip recording a participant assembling one of the three plates of the designed Generic Assembly Box. Each instance consists of two-level temporal annotations: primitive task and atomic action, and spatial annotations, which means the bounding boxes for subjects, objects, and tools.

(b) How many instances are there in total?

We recorded 3222 videos over 86.9 hours, totaling over 1.5M frames. To ensure annotation quality, we manually labeled temporal annotations for 609 plate assembly videos and spatial annotations for over 144K frames.

(c) Does the dataset contain all possible instances, or is it a sample (not necessarily random) of instances from a larger set?

Yes, the dataset contains all possible instances.

(d) What data does each instance consist of?

See 2. (a).

(e) Is there a label or target associated with each instance?

See 2. (a).

(f) Is any information missing from individual instances?

No.

(g) Are relationships between individual instances made explicit?

Yes, each instance (video clip) contains one participant performing one task (assembling one of the three plates of the designed Generic Assembly Box.)

(h) Are there recommended data splits?

For action recognition and action segmentations, we provide two data splits: trainset and testset.

For object detection and multi-object tracking, we provide another two data splits: trainset and testset.

Refer to Section 2.4 for details.

(i) Are there any errors, sources of noise, or redundancies in the dataset?

Given the scale of the dataset and complexity in annotation, it is possible that some ad-hoc errors exist in our annotations. However, we have given our best efforts (via human checks and quality checking code scripts) in examining manually labelled annotations to minimize these errors.

(j) Is the dataset self-contained, or does it link to or otherwise rely on external resources (e.g., websites, tweets, other datasets)?

The dataset is self-contained.

(k) Does the dataset contain data that might be considered confidential (e.g., data that is protected by legal privilege or by doctor-patient confidentiality, data that includes the content of individuals' non-public communications)?

No.

(l) Does the dataset contain data that, if viewed directly, might be offensive, insulting, threatening, or might otherwise cause anxiety?

No.

(m) Does the dataset relate to people?

Yes, all videos are recordings of human assembly activities, and all annotations are related to the activities.

(n) Does the dataset identify any subpopulations (e.g., by age, gender)?

No. Our participants have different ages and genders. But our dataset does not identify this information. To ensure this, we have blurred participants' faces in the released videos.

(o) Is it possible to identify individuals (i.e., one or more natural persons), either directly or indirectly (i.e., in combination with other data) from the dataset?

No, as explained in 2. (n), we have blurred participants' faces in the released videos.

(p) Does the dataset contain data that might be considered sensitive in any way (e.g., data that reveals racial or ethnic origins, sexual orientations, religious beliefs, political opinions or union memberships, or locations; financial or health data; biometric or genetic data; forms of government identification, such as social security numbers; criminal history)?

No.

(q) Any other comments?

None.

3. Collection Process

(a) How was the data associated with each instance acquired?

For each video instance, we provide temporal annotations and spatial annotations. We follow HR-SAT to create temporal annotations to ensure the annotation consistency. The temporal annotations were manually created and checked by our researchers. The spatial annotations were manually created by postgraduate students at the University of Auckland, who were trained by one of our researchers to ensure the annotation quality.

(b) What mechanisms or procedures were used to collect the data (e.g., hardware apparatus or sensor, manual human curation, software program, software API)?

Data were collected on three Azure Kinect RGB+D cameras via live video capturing while a participant is performing the assembly actions, and we manually labeled all the annotations.

(c) If the dataset is a sample from a larger set, what was the sampling strategy (e.g., deterministic, probabilistic with specific sampling probabilities)?

No, we created a new dataset.

(d) Who was involved in the data collection process (e.g., students, crowdworkers, contractors) and how were they compensated (e.g., how much were crowdworkers paid)?

For video recordings, volunteer participants were rewarded gift cards worth NZ$50.00 upon completion of the 2-hour data collection session.

For data annotations, we contracted students at the University of Auckland, and they were paid at a rate of NZ$23.00 per hour.

(e) Over what timeframe was the data collected?

The videos were recorded during August to September of 2022, and the annotations were made during October of 2022 to March of 2023.

(f) Were any ethical review processes conducted (e.g., by an institutional review board)?

Yes, we obtained ethics approval from the University of Auckland Human Participants Ethics Committee. More information can be found in Section 5.

(g) Does the dataset relate to people?

Yes, we recorded the process of people assembling the Generic Assembly Box.

(h) Did you collect the data from the individuals in question directly, or obtain it via third parties or other sources (e.g., websites)?

We collected the data from the individuals in question directly.

(i) Were the individuals in question notified about the data collection?

Yes, all participants were informed of the data collection purpose, process and the intended use of the data. They were sent a Participant Information Sheet and signed Consent Form prior to the collection session. All sessions started with an introduction where instructions on data collection, health and safety and confirmation of the Consent Form were discussed.

(j) Did the individuals in question consent to the collection and use of their data?

Yes, all participants were sent a Participant Information Sheet and Consent Form prior to the collection session. We confirmed that they had agreed to and signed the Consent form regarding the collection and use of their data before proceeding with any data collection. Details can be found in Section 5.

(k) If consent was obtained, were the consenting individuals provided with a mechanism to revoke their consent in the future or for certain uses?

Yes. The Participant Information Sheet and Consent Form addressed how they can request to withdraw and remove their data from the project and how the data will be used.

(l) Has an analysis of the potential impact of the dataset and its use on data subjects (e.g., a data protection impact analysis) been conducted?

No, all data have been processed to be made de-identifiable and all annotations are on objective world states. The potential impact of the dataset and its use on data subjects were addressed in the Ethics Approval, Participant Information Sheet and Consent Form. Details can be found in Section 5.

(m) Any other comments?

None.

4. Preprocessing, Cleaning and Labeling

(a) Was any preprocessing/cleaning/labeling of the data done (e.g., discretization or bucketing, tokenization, part-of-speech tagging, SIFT feature extraction, removal of instances, processing of missing values)?

Yes, we have cleaned the videos by blurring participants' faces. We have also extracted I3D features from the video for action segmentation benchmarking.

(b) Was the "raw" data saved in addition to the preprocessed/cleaned/labeled data (e.g., to support unanticipated future uses)?

No, we only provide the cleaned videos (participants' faces being blurred) to the public due to the ethics issues.

(c) Is the software used to preprocess/clean/label the instances available?

Yes, we used CVAT to draw bounding boxes. Details can be found in Section 2.3.

(d) Any other comments?

None.

5. Uses

(a) Has the dataset been used for any tasks already?

No, the dataset is newly proposed by us.

(b) Is there a repository that links to any or all papers or systems that use the dataset?

Yes, we provide the link to all related information on our website.

(c) What (other) tasks could the dataset be used for?

The dataset can also be used for Compositional Action Recognition, Human-Object Interaction Detection, and Visual Question Answering.

(d) Is there anything about the composition of the dataset or the way it was collected and preprocessed/cleaned/labeled that might impact future uses?

We granulated the assembly action annotation into subject, action verb, manipulated object, target object and tool. We believe the fine-grained and compositional annotations can be used for more detailed and precise descriptions of the assembly process, and the descriptions can serve various real-world industrial applications, such as robot learning, human robot collaboration, and quality assurance.

(e) Are there tasks for which the dataset should not be used?

The usage of this dataset should be limited to the scope of assembly activity or task understanding, e.g., action recognition, action segmentation, action anticipation, human-object interaction detection, visual question answering, and the downstream industrial applications, e.g., robot learning, human-robot collaboration, and quality assurance. Any work that violates our Code of Conduct are forbidden. Code of Conduct can be found at our website[11].

(f) Any other comments?

None.

6. Distribution

(a) Will the dataset be distributed to third parties outside of the entity (e.g., company, institution, organization) on behalf of which the dataset was created?

Yes, the dataset will be made publicly available.

(b) How will the dataset will be distributed (e.g., tarball on website, API, GitHub)?

The dataset could be accessed on our website.

(c) When will the dataset be distributed?

We provide private links for the review process. Then the dataset will be released to the public after the review process.

(d) Will the dataset be distributed under a copyright or other intellectual property (IP) license, and/or under applicable terms of use (ToU)?

We release our dataset and benchmark under CC BY-NC 4.0[12] license.

(e) Have any third parties imposed IP-based or other restrictions on the data associated with the instances?

No.

(f) Do any export controls or other regulatory restrictions apply to the dataset or to individual instances?

---

[11]https://iai-hrc.github.io/ha-vid
[12]https://creativecommons.org/licenses/by-nc/4.0/

No.

(g) Any other comments?

None.

7. Maintenance

(a) Who is supporting/hosting/maintaining the dataset?

Regina Lee and Hao Zheng are maintaining, with continued support from Industrial AI Research Group at The University of Auckland.

(b) How can the owner/curator/manager of the dataset be contacted (e.g., email address)?

E-mail addresses are at the top of the paper.

(c) Is there an erratum?

Currently, no. As errors are encountered, future versions of the dataset may be released and updated on our website.

(d) Will the dataset be updated (e.g., to correct labeling errors, add new instances, delete instances')? Yes, see 7.(c).

(e) If the dataset relates to people, are there applicable limits on the retention of the data associated with the instances (e.g., were individuals in question told that their data would be retained for a fixed period of time and then deleted)?

No.

(f) Will older versions of the dataset continue to be supported/hosted/maintained?

Yes, older versions of the dataset and benchmark will be maintained on our website.

(g) If others want to extend/augment/build on/contribute to the dataset, is there a mechanism for them to do so?

Yes, errors may be submitted to us through email.

(h) Any other comments?

None.