# OpenReview forum: "HA-ViD: A Human Assembly Video Dataset for Comprehensive Assembly Knowledge Understanding"
_NeurIPS.cc/2023/Track/Datasets_and_Benchmarks — NeurIPS 2023 Datasets and Benchmarks Poster_

### Official Review · Reviewer_soUy · 2023-07-15
**Reviews for 760**

**Rating:** 7
**Confidence:** 3
**Correctness:** Yes.

**Strengths:**

1. The dataset is challenging and interesting.
2. Author has carely considered industrial assembly, and presented some usefull designs.
3. The dataset can support multiple tasks.

**Additional Feedback:**

See the above.

**Clarity:**

More details about assembly which is rare in daily life can be illustrated for better understand.

**Documentation:**

Yes.

**Limitations:**

They have discussed in short in the conclusion part.

**Opportunities For Improvement:**

1. Detailed comparison with exsiting datasets, including quantity, tasks.
2. Latest methods in action recognition and segmentation should be tested. I find most methods are before 2020.
3.  How to achieve understanding or evaluate the assembly quaility?

**Relation To Prior Work:**

It has been compared in Tab.1.

**Summary And Contributions:**

This paper provide an assembly understanding benchmark which includes action recognition, action segmentation, object detection, and object tracking. It seems the first time annoting human pauses and errors for comprehensive knowledge understanding.

---

> ### Author Response · Authors · 2023-08-21
> **Response to Reviewer soUy**
>
> Thank you for your recognition on the contributions of our work and the positive feedback on the challenges, utility, and multi-task support of our dataset.
>
> > *"Detailed comparison with exsiting datasets, including quantity, tasks."*
>
> We agree that quantitative comparison between our dataset and others would emphasize our dataset’s features. Therefore, we have added a new table in the Supplementary (Line 74 and Table 2) to compare our dataset with others on various parameters, including subjects, videos, views, unique sequences, total duration, average clip length, temporal label classes and spatial label classes. Regarding the tasks, we compare the tasks on our dataset with existing datasets in Line 157 and Line 194 of the Manuscript.
>
> > *"Latest methods in action recognition and segmentation should be tested. I find most methods are before 2020."*
>
> Thank you for your suggestion and we agree. In the revision, in addition to the algorithms we originally chose, we benchmarked the recent state-of-the-art algorithms: ST-GCN++ (Duan, et al., 2022), UniFormerV2 (Li, et al., 2022), and C2F-TCN (Singhania, et al., 2023) (updated in Line 137). The benchmarking results can be found in the updated Table 2 in the Manuscript and the details can be found in the Supplementary (Section 3.1 (Line 123, Line 135 and Table 3, Table 4) and Section 3.2 (Line 161 and Table 5, Table 6)).
>
> > *"How to achieve understanding or evaluate the assembly quaility?"*
>
> Thank you for your thoughtful question on understanding and evaluating assembly quality. Assembly quality can be understood as whether the product was assembled correctly. One way to evaluate this is to identify the presence of wrong actions—actions performed in the wrong way or order. Another way to evaluate this is to identify incorrectly assembled parts using 3D and pixel-wise geometric information. We have enhanced our discussion of Insight #4 regarding wrong action annotations (Line 208) and added a discussion on our plans to expand HA-ViD to include the 3D and pixel-wise geometric information for assembly quality checking (Line 254).

---

### Official Review · Reviewer_NUWb · 2023-07-28
**Review for "HA-ViD: A Human Assembly Video Dataset for Comprehensive Assembly Knowledge Understanding"**

**Rating:** 8
**Confidence:** 4
**Correctness:** The conclusions are well-supported by…
**Clarity:** This paper is well-writen and easy to…

**Strengths:**

1. This article introduces a well-integrated dataset with assembly scenarios, demonstrating significant practical value. It offers four distinct types of annotations that support a variety of video understanding tasks."

2. The paper provides a comprehensive showcase of the dataset's data distribution and statistical characteristics, making it easy to understand the dataset's purpose.

3. Through the analysis of experimental results, the authors offer valuable insights into the dataset's potential applications and future directions.

**Additional Feedback:**

I have no more comment.

**Documentation:**

I have no concern about this part.

**Ethics:**

No.

**Limitations:**

The authors have adequately addressed the limitations.

**Opportunities For Improvement:**

1. Human-object interaction detection or scene graph generation would also hold significant value in this context, as they can integrate human actions with specific components, providing a more comprehensive understanding.

2. For each task, the authors could consider making preliminary attempts with classical methods to solve the proposed challenges. For instance, tackling imbalanced data issues by trying balanced sampling or focal loss, and analyzing the effectiveness of these classical approaches.

**Relation To Prior Work:**

Yes.

**Summary And Contributions:**

The paper introduces HA-ViD, a human assembly video dataset designed for futuristic ultra-intelligent industries, which contains representative assembly scenarios. The authors provides detailed annotation to support four fundamental video understanding tasks: action recognition, action segmentation, object detection and multi-object tracking.

---

> ### Author Response · Authors · 2023-08-21
> **Response to Reviewer NUWb**
>
> Thank you for reviewing our work and providing positive feedback on the dataset practicality, clarity of dataset purpose, and insights. We appreciate your acknowledgement of the contributions of our dataset.
>
> > *"Human-object interaction detection or scene graph generation would also hold significant value in this context, as they can integrate human actions with specific components, providing a more comprehensive understanding."*
>
> Thank you for your suggestion to consider human-object interaction or scene graph generation tasks. Although our current work focuses on the foundational video understanding tasks, video reasoning tasks are essential for comprehensive understanding of assembly knowledge. Understanding human-object interaction can provide important insights into how humans interact with objects and enable finer-grained tracking of the assembly process. Scene graph generation is also valuable to understand the context of an assembly task as it can capture the relationship between the elements in the assembly scene such as the action, objects, subjects, and tools. We have extended our Insight discussions to discuss these aspects in understanding assembly progress (Line 173), process efficiency (Line 199) and learning assembly skills (Line 234).
>
> > *"For each task, the authors could consider making preliminary attempts with classical methods to solve the proposed challenges. For instance, tackling imbalanced data issues by trying balanced sampling or focal loss, and analyzing the effectiveness of these classical approaches."*
>
> Thank you for your suggestion to make preliminary attempts to solve challenges such as the imbalanced data issue. Analyzing the effectiveness of preliminary solutions can enhance the insights we provide, especially for Insight #3, which discusses the imbalanced data problem. Aligned with your suggestions, we explore the effectiveness of randomly over sampling the long tail classes in tackling the imbalanced data problem. As a result, we reported our preliminary results in Line 204 and discuss this in the Supplementary (Line 259), to inspire future research on our dataset on this aspect.

---

### Official Review · Reviewer_VsBS · 2023-07-28
**Review of HA-ViD**

**Rating:** 6
**Confidence:** 4
**Correctness:** Yes.
**Clarity:** Well.

**Strengths:**

+ The assembly understanding matters for many studies, as the authors discussed. A large and well-annotated dataset contributes a lot.

+ Detailed introductions and analyses are given to better understand the data and task.

+ The whole paper is written well and easy to follow.

+ Essential resources are given.

**Additional Feedback:**

1. Fig. 3: the front is too small.

2. L97: progression.Our annotations --> progression. Our annotations (missing a space

3. Suppl. Fig. 12: please adjust the y-axis.

4. Some scores in the Suppl. Tables can be bold to be easier to read.

**Documentation:**

Good and detailed documentation is given in the supplementary and website.

**Ethics:**

Well discussed in the suppl.

**Limitations:**

- The task setting: for the assembly task, the action understanding and low-level object task are important, but the assembly instruction reasoning and planning are more vital for a new dataset. There are already many datasets for the provided four tasks. Thus, I suggest the authors redesign the benchmark and propose more reasoning and planning tasks based on the data. This would benefit the finer and more complex action understanding, robot skill learning, human-robot cooperation, etc. For example, next-step prediction, step inpainting, instruction VQA, step order arrangement, etc.

-  The annotation: lacking a very important kind of information for assembly data, i.e., human hand pose. As many actions in the dataset like insert, rotate, etc. At least, estimated and human-adjusted hand 3D key points can be provided. Besides, the Kinetic based 3D pose estimation may be often inaccurate, this should be considered and analyzed.

- Given multi-view videos, readers may want to see more tasks of 3D or 4D reconstructions, hand-object contact detection, etc.

**Opportunities For Improvement:**

- Maybe the knowledge transfer experiment from human demo to robot imitation in the syn engine would make the contribution stronger.

- The discussed tasks are all low and mid-level, lacking reasoning and planning tasks. Besides, the two-hand cooperation also can provide many interesting possibilities.

- More analyses (figures, tables) would be better to introduce the data properties, e.g., annotators, participants, lighting, instructions, and camera views.

- Given the wrong action annotations, a discussion about the re-planning would be better.


**Relation To Prior Work:**

Yes, clear.

**Summary And Contributions:**

This paper proposes a new dataset of object assembly with multi-modal and multi-view data. A large-scale of videos are collected and annotated with temporal and spatial labels for action recognition, segmentation, object detection, and multi-object tracking. Several SOTA methods are evaluated on this new benchmark with detailed analysis. This dataset proposes a meaningful and valuable data contribution to the community. However, several task and annotation problems exist.

---

> ### Author Response · Authors · 2023-08-21
> **Response to Reviewer VsBS (Part 1 of 2)**
>
> Thank you for your valuable comments and recognition. We acknowledge the concerns raised regarding certain tasks and annotations and have addressed them in the following responses.
>
> > *"Maybe the knowledge transfer experiment from human demo to robot imitation in the syn engine would make the contribution stronger."*
>
> We agree the experiment of robot imitation learning in a synthetic engine would effectively demonstrate how our dataset supports human-robot skill transfer. This process involves learning skill parameters from our dataset and using them as input in a robot learning environment, such as a synthetic engine. We have added discussions on how our dataset can support skill parameter learning (Line 234) and its limitations as possible research the community could work on using our dataset (Line 254). Since the primary objective of our work is to present HA-ViD, focusing on video understanding and reasoning tasks for comprehending assembly knowledge, our current benchmarks do not include the suggested task.
>
> > *"The discussed tasks are all low and mid-level, lacking reasoning and planning tasks. Besides, the two-hand cooperation also can provide many interesting possibilities ...The task setting: for the assembly task, the action understanding and low-level object task are important, but the assembly instruction reasoning and planning are more vital for a new dataset. There are already many datasets for the provided four tasks. Thus, I suggest the authors redesign the benchmark and propose more reasoning and planning tasks based on the data. This would benefit the finer and more complex action understanding, robot skill learning, human-robot cooperation, etc. For example, next-step prediction, step inpainting, instruction VQA, step order arrangement, etc."*
>
> Thank you for your insightful feedback. We agree that high level reasoning tasks are crucial for deeper understanding of assembly knowledge. To highlight this point throughout our manuscript, we have taken the following steps:
>
> 1. We address reasoning in our introduction and Figure 1
> 2. We have enhanced our insights in Sections 3.2 to 3.5 to further discuss how reasoning tasks will enable us to take the next step from foundational video understanding, towards acquiring application-oriented assembly knowledge. In Line 216, we discuss the interesting research towards understanding two-hand cooperation.
> 3. We additionally benchmarked FUTR (Gong, et al., 2022), a SOTA long-term anticipation method, to take a step towards assembly progress reasoning (Line 173 in the Manuscript and Line 238 in the Supplementary).
> 4. We added a Discussion Section to discuss how our dataset may be extended to further support reasoning tasks for acquiring real-world assembly knowledge (Line 254).
>
> > *"More analyses (figures, tables) would be better to introduce the data properties, e.g., annotators, participants, lighting, instructions, and camera views."*
>
> Thank you for the suggestion to improve the way we introduce the data properties of our dataset. We agree that a visual illustration would improve our introduction of the data properties relating to the data collection setup. Following your suggestion, we added a new figure in the Supplementary (Line 40 and Figure 6) to better illustrate the participants, lighting, instructions, and camera views.
>
> > *"Given the wrong action annotations, a discussion about the re-planning would be better."*
>
> Thank you for the valuable suggestion. We fully agree that our “wrong” annotations can be useful to understand task re-planning. Given our “wrong” annotations, the subsequent actions performed by the participants can be analyzed to understand and learn how humans re-plan tasks after making a mistake. To address your comment, we revised the discussion of Insight #4 in the Manuscript (Line 208) to elaborate the three key research directions that wrong action annotations enable, including re-planning.

---

> > ### Author Response · Authors · 2023-08-21
> > **Response to Reviewer VsBS (Part 2 of 2)**
> >
> > > *"The annotation: lacking a very important kind of information for assembly data, i.e., human hand pose. As many actions in the dataset like insert, rotate, etc. At least, estimated and human-adjusted hand 3D key points can be provided. Besides, the Kinetic based 3D pose estimation may be often inaccurate, this should be considered and analyzed."*
> >
> > Given multi-view videos, readers may want to see more tasks of 3D or 4D reconstructions, hand-object contact detection, etc.
> > Thank you for your comments regarding hand pose and 3D key point data. We agree that hand pose and 3D key point annotations can provide valuable information in acquiring knowledge for learning assembly skills, understanding human motion, and product quality checking. Having these annotations would enable a wider range of tasks such as 3D or 4D reconstructions and hand-object contact detection. Therefore, in our added discussion section, we discuss our thoughts on how our dataset can be extended to capture these annotations (Line 254).
> >
> > We also acknowledge the skeleton positions we use are not the ground truth and that the precision of the input skeleton is important for skeleton-based recognition methods. This is a limitation of the skeleton modality itself. Ground truth labels require intricate motion capture using sensors and markers which was not ideal to have for our data collection setup. Therefore, we chose the Azure Kinect Body Tracking SDK as one of the best available skeleton extraction methods. We fully agree with your comment that this is an important point to discuss regarding skeleton-based action recognition and have extended our discussion in the Supplementary (Line 89 and Line 214).
> >
> > >1. *"Fig. 3: the front is too small."*
> > >2. *"L97: progression.Our annotations --> progression. Our annotations (missing a space)"*
> > >3. *"Suppl. Fig. 12: please adjust the y-axis."*
> > >4. *"Some scores in the Suppl. Tables can be bold to be easier to read."*
> >
> > Thank you for the bringing these issues to our attention. We have made the following revisions to address this:
> > 1. The font size of Figure 3 has been increased.
> > 2. We have fixed the spacing of Line 97.
> > 3. We have adjusted the y-axis of Supplementary Figure 12 to better show the distribution of the long-tail classes.
> > 4. We have bolded some scores of the benchmark tables in the Supplementary (Table 3, 5, 9, 10) to highlight the best benchmark performances.

---

> > ### Comment · Reviewer_VsBS · 2023-08-21
> > **Post-rebuttal**
> >
> > Thanks for the response. I appreciate the new additional content.  My main concern is the high-level reasoning tasks based on this dataset which would benefit many downstream tasks a lot. Hope the authors can keep maintaining their benchmark and provide more tasks and tracks for robot learning and multi-modal reasoning.
> >
> > Overall, I think this work is useful and can inspire future work.

---

### Official Review · Reviewer_ZuD2 · 2023-08-02
**Review of HA-ViD**

**Rating:** 7
**Confidence:** 3
**Correctness:** Yes, this dataset is built soundly.
**Clarity:** Yes.

**Strengths:**

1. This dataset is well-motivated. Since assembly knowledge understanding from videos is crucial for futuristic ultra-intelligent industrial applications, the lack of effective data will hinder the development of this field.
2. This dataset is well designed from the massive captured data, the applicable scenarios, and comprehensive annotations. Compared with previous assembly video datasets, this dataset has a clear superiority. The whole pipeline is sound.
3. The processes of dataset collection and annotation are clear and informative.
4. The experiments provide four benchmarks on different tasks. The provided baseline methods are representative, like the action recognition benchmark, using three kinds of methods from five models. In the appendix, most of the experimental details are given.

**Additional Feedback:**

Please see "Opportunities For Improvement".

**Documentation:**

Yes, the dataset URL is provided.

**Ethics:**

No.

**Limitations:**

Yes, the main paper has discussed the limitations.

**Opportunities For Improvement:**

1. This dataset is claimed to be a multi-modality dataset (#L9 and the caption of Figure 1). From the main paper, it seems to mean each video contains one task. However, from the website video, it means multi-modalities from Kinect (RGB, depth, skeleton). I'm not sure about whether it is proper. In addition, I wonder why the main paper did not introduce the use of depth maps and skeleton sequences. Are they not effective modalities for assembly knowledge understanding or the mentioned four fundamental tasks? Another minor question is about the use of ST-GCN in the action recognition benchmark because of the bad performance. In Line 84 of the Appendix ("we first extracted the upper 26 skeleton joints from each frame”), where do these 26 skeleton joints come from? How to make sure the precision of the input skeleton positions?
2. Since this dataset focuses on assembly knowledge understanding, the hand-object-interaction estimation (e.g., via a MANO model and object meshes), hand-object mesh contact estimation, and hand pose generation from the given language guidance are also important. Please discuss the essence of these tasks. These 3D annotations will be more complex, and it will be hard to do manual annotations. If so, how to handle these issues may be discussed in future work.
3. From Table 1, some quantitative statistics comparisons of existing datasets should be conducted (maybe in another table), such as the frames, subjects, actions, and clip numbers, to highlight the superiority of the proposed dataset.



**Relation To Prior Work:**

Yes.

**Summary And Contributions:**

HA-ViD is a human assembly video dataset for advancing comprehensive assembly knowledge understanding of real-world industrial applications. It designs several assembly scenarios and a three-stage progressive learning setup to capture the natural process of human procedural knowledge acquisition. For annotations, it provides multi-view, multi-modality data, fine-grained action annotations, human pause and error annotations, and collaboration status annotations. Several annotation pipelines are introduced. From experiments, this dataset benchmarks four foundational tasks and provides some key insights into application-oriented knowledge.

---

> ### Author Response · Authors · 2023-08-21
> **Response to Reviewer ZuD2**
>
> Thank you for your comments and positive feedback on our motivation, design, annotation process and benchmarking. We appreciate your recognition of our work.
>
> > *"This dataset is claimed to be a multi-modality dataset (#L9 and the caption of Figure 1). From the main paper, it seems to mean each video contains one task. However, from the website video, it means multi-modalities from Kinect (RGB, depth, skeleton)."*
>
> Thank you for identifying this issue and we apologize for any confusion caused. We claim to have a multi-modality dataset because we provide multi-modalities of data (RGB, depth, skeleton). We mention “each video contains one task” to explain what assembly process is captured in each video (as in Line 108). However, we notice the segment, "…multi-modality videos (each video contains one assembly task) …" is confusing in the Abstract and Figure 1 caption. We have removed this bracket content to address this issue.
>
> > *"I wonder why the main paper did not introduce the use of depth maps and skeleton sequences. Are they not effective modalities for assembly knowledge understanding or the mentioned four fundamental tasks?"*
>
> Thank you for your comment. We agree that these modalities are useful for assembly knowledge understanding. Skeleton data is an efficient modality that can be used to understand human motion. We use skeleton data to benchmark ST-GCN, a skeleton-based action recognition. Also, in our revision, we additionally benchmark ST-GCN++ (Duan, et al., 2022) on HA-ViD (Line 140 and Table2). Depth data can be used to obtain pose information of parts, tools and subjects. Although depth data is not typically used as input for the four fundamental tasks, we believe our provided depth modality can be used when taking the next step in assembly process comprehension, toward 3D reasoning. We have added this discussion in Line 234 and Line 254.
>
> > *"… where do these 26 skeleton joints come from? How to make sure the precision of the input skeleton positions?"*
>
> Thank you for your questions regarding our ST-GCN benchmark. We extracted the skeleton data via the Azure Kinect Body Tracking SDK. We use the upper 26 joints from the total 32 joints extracted by the SDK. We exclude the lower limb joints (left and right knee, ankle, and foot joints) that cannot be seen due to the participant standing behind the workbench. We have revised the Supplementary (Line 89) to elaborate this point.
>
> We acknowledge the skeleton positions we use are not the ground truth and that the precision of the input skeleton is important for skeleton-based recognition methods. This is a limitation of the skeleton modality itself. Ground truth labels require intricate motion capture using sensors and markers which was not ideal to have for our data collection setup. Therefore, we chose the Azure Kinect Body Tracking SDK, one of the best skeleton extraction methods. We fully agree with your comment that it is important to consider the skeleton data quality and have added a discussion in the Supplementary (Line 214).
>
> > *"… hand-object-interaction estimation … hand-object mesh contact estimation, and hand pose generation from the given language guidance are also important. Please discuss the essence of these tasks. These 3D annotations will be more complex … how to handle these issues may be discussed in future work."*
>
> Thank you for your suggestion to discuss assembly knowledge understanding tasks such as hand-object interaction, hand-object contact estimation, and hand pose generation. We agree that these tasks can reveal how humans manipulate and interact with objects during assembly, which affects the assembly efficiency, assembly quality, and ergonomics. Therefore, they are essential in real-world applications involving complex two-handed collaboration and manipulation. We integrate the abovementioned tasks in our Insights discussion about understanding assembly progress (Line 173), process efficiency (Line 200), two-handed collaboration (Line 223), and learning assembly skills (Line 234).
>
> We also recognize that these tasks will require more detailed 3D annotations. Due to the challenges in manually annotate the 3D annotations, future work on our dataset could involve incorporating additional sensors and markers to track interested 3D points to form various 3D annotations. Therefore, we have added a discussion (Line 254) with thoughts on how our dataset may be extended to capture these annotations.
>
> > *"…some quantitative statistics comparisons of existing datasets should be conducted…"*
>
> We agree this comparison would emphasize the dataset’s features. Accordingly, we’ve added a new table in the Supplementary (Line 74, Table 2) to compare our dataset with others on various parameters, including subjects, videos, views, unique sequences, total duration, average clip length, temporal label classes and spatial label classes.

---

> > ### Comment · Reviewer_ZuD2 · 2023-08-30
> > **Reply**
> >
> > Thanks for the authors' replies. For the answer to "… where do these 26 skeleton joints come from? How to make sure the precision of the input skeleton positions?", I think that using the poses from Kinect is not a good choice since existing human pose estimation algorithms have obtained better performance. There are some useful works to help the annotation: (1) 2D whole-body keypoint detector (DWPose[1]) to detect the hand, face, and body keypoints; (2) 3D SMPL-X estimator with a good handle on hand interaction (InterWild[2]); (3) SOTA SMPL-X estimator with large-scale training datasets (SMPLer-X[3] and OSX[4]).
> >
> > Based on the above modification, I will keep my score.
> >
> > Reference:
> > [1] Effective Whole-body Pose Estimation with Two-stages Distillation, ICCVW 2023. https://github.com/IDEA-Research/DWPose
> > [2] Bringing Inputs to Shared Domains for 3D Interacting Hands Recovery in the Wild, CVPR 2023. https://github.com/facebookresearch/InterWild
> > [3] https://github.com/caizhongang/SMPLer-X
> > [4] One-Stage 3D Whole-Body Mesh Recovery with Component Aware Transformer, CVPR 2023. https://github.com/IDEA-Research/OSX

---

### Official Review · Reviewer_YGBP · 2023-08-03
**HA-ViD comments**

**Rating:** 7
**Confidence:** 4
**Correctness:** Yes
**Clarity:** Yes

**Strengths:**

1. This work is the first human assembly video dataset for comprehensive understanding of assembly knowledge.
2. This topic is very interesting and the dataset is completed.

**Additional Feedback:**

NA

**Documentation:**

Yes

**Limitations:**

Yes

**Opportunities For Improvement:**

More data is need in this area, so I wish this dataset can be expented with more data and more diverse scenes in the future.

**Relation To Prior Work:**

Yes

**Summary And Contributions:**

This study presents HA-ViD,  human assembly video dataset. The dataset contains representative industrial assembly scenarios, a natural procedural knowledge acquisition process, and consistent human-machine shared annotations. By providing 3222 multi-view, multi-modal videos, 1.5M frames, 96K temporal labels and 2M spatial labels, this work evaluate four basic video understanding tasks and analyze their performance in understanding assembly progress, process efficiency, Performance in terms of task collaboration, skill parameters, and human intention.

---

> ### Author Response · Authors · 2023-08-21
> **Response to Reviewer YGBP**
>
> Thank you for your comments and positive feedback on the novelty and completeness of our dataset. We appreciate your recognition of our knowledge contributions.
>
> > *"More data is need in this area, so I wish this dataset can be expented with more data and more diverse scenes in the future."*
>
> Thank you for your inspiring suggestion. We fully agree that expanding HA-ViD to include more diverse scenes is an important step toward developing robust techniques for solving futuristic industrial assembly problems. Our annotations, that strictly adhere to HR-SAT, provides a foundation for expanding HA-ViD in a consistent manner. We have elaborated this point in Line 92 and Line 246. We have also added a discussion on the possible expansion directions in Line 250 and Line 254. Furthermore, we have added a Roadmap on the Github repository and will use Github versioning to manage future releases from us and followers. The Github Repository is available at: https://github.com/iai-hrc/ha-vid

---

### Official Review · Reviewer_ei2t · 2023-08-05
**Review of HA-ViD**

**Rating:** 6
**Confidence:** 4
**Correctness:** Yes.
**Clarity:** Yes.

**Strengths:**

HA-ViD is the first one focusing on human assembly videos.
It captures wide-ranging assembly knowledge, including extensive multi-view videos, frames, and labels.
This dataset has potential for research on next-generation intelligent industry applications.

**Additional Feedback:**

N/A

**Documentation:**

It seems to be great from the attached links.

**Ethics:**

No.

**Limitations:**

Yes, L233-235.

**Opportunities For Improvement:**

The dataset requires a wider range of tasks and scenes.
The dataset/benchmark could be more fine-grained, for example, including tasks that require 3D reasoning and two-hand cooperation.
It would be great to have a more detailed discussion on how to effectively scale and utilize this dataset in real-world applications.

**Relation To Prior Work:**

Yes.

**Summary And Contributions:**

The paper introduces the HA-ViD dataset, the first human assembly video dataset capturing comprehensive assembly knowledge. It includes industrial assembly scenarios, procedural knowledge acquisition, and human-robot shared annotations. The dataset is benchmarked for action recognition, action segmentation, object detection, and multi-object tracking, aiming to advance research in assembly knowledge understanding for future intelligent industry applications.

---

> ### Author Response · Authors · 2023-08-21
> **Response to Reviewer ei2t**
>
> Thank you for taking the time to review our manuscript and provide valuable feedback. We appreciate your positive feedback on the potential of our dataset. We have addressed your concerns as below.
>
>
> > *"The dataset requires a wider range of tasks and scenes."*
>
> Thank you for your suggestion and we agree that industrial assembly applications feature diverse and complex operation tasks in various scenes. To ensure our dataset is representative, we (1) designed our assembly product to have a wide range of parts, including those common in industrial assembly, e.g., screws, nuts, and bolts; (2) designed the assembly operations to feature complex task precedence and coordination of both hands on a mixture of assembly parts and tools; and (3) designed assembly settings featuring the participants following different instructions. By doing so, we expect our dataset can capture the essence of real-world assembly settings.
> However, recognizing the need of more real-world assembly datasets to understand human assembly knowledge, we added a discussion section with thoughts on how our dataset may be extended systematically to cover new tasks and scenes (Line 250 and Line 254).
>
>
> > *"The dataset/benchmark could be more fine-grained, for example, including tasks that require 3D reasoning and two-hand cooperation."*
>
> Thank you for the suggestion on two-hand cooperation and 3D reasoning tasks. We recognize the significance of understanding two-hand collaboration and 3D reasoning in various real-world applications, such as assembly progress monitoring and human-robot collaboration. Our dataset aims to support tasks for two-hand cooperation understanding by providing separate two-hand annotations and inter-hand collaboration status annotations (Line 97). In Line 216, we have elaborated on how our dataset can support tasks that focus on two-hand collaboration. In Section 3.4, Line 216, we have elaborated on how our dataset can support tasks that focus on two-hand collaboration. In Line 234, we extended our discussion on how 3D reasoning tasks can enable skill parameters acquisition. In Section 3.5, Line 234, we extended our insight to discuss how 3D reasoning tasks can enable skill parameters acquisition. For 3D reasoning, we acknowledge our dataset would benefit from future work providing 3D key point annotations. To address this, we added a discussion section, outlining the systematic approach to extend HA-ViD (Line 254).
>
>
> > *"It would be great to have a more detailed discussion on how to effectively scale and utilize this dataset in real-world applications."*
>
> Inspired by your insights, we have decided to add discussions on the practical use and scalability of our dataset in real-world applications (Section 4 Line 243). Our key thoughts are as below:
>
> HA-ViD provides a basis for training video understanding and reasoning tasks for understanding assembly sequence, efficiency, collaboration dynamics and human-object interactions. These knowledges, at a foundational level, can be leveraged to bootstrap knowledge understanding for the target application-specific scenario via transfer learning, for example. Our systematic annotation following HR-SAT also provides the community with methods to annotate and add new application-specific data to HA-ViD. This allows researchers to reuse HA-ViD, together with potentially a small target dataset, to develop application-specific models.

---

> > ### Comment · Reviewer_ei2t · 2023-08-22
> >
> > Thank you for your response. Most of my concerns have been addressed, so I will keep my positive reviews. I would like to see more potential of this paper for real-world applications in future research.

---

### Official Review · Reviewer_HHfr · 2023-08-07
**Good, Useful, Interesting, Novel Assembly Video Dataset**

**Rating:** 7
**Confidence:** 5
**Correctness:** Overclaims and use of superlatives sh…
**Clarity:** Yes, but writing can be improved as s…

**Strengths:**

This paper is good and would be useful to community as it contains hierarchical structure, detailed annotations about multi-level task, bboxes around hand, tools, parts, etc. Dataset also covers progressive proficiency in assembly tasks.

**Additional Feedback:**

Good paper with good potential, I would suggest not to make it another "sell out".

**Documentation:**

Yes.

**Ethics:**

No.

**Limitations:**

Yes.

**Opportunities For Improvement:**

1. overclaims and use of superlatives should be adjusted as mentioned in summary.
2. L199 this insight is not clear. Can authors please elaborate?

**Relation To Prior Work:**

Yes, but overclaims kind of do not respect work already done previously by prior efforts. This should be adjusted. Credit should be given where it is due.

**Summary And Contributions:**

This paper introduces HA-ViD dataset to support automated video understanding of assembly line-related tasks such as skill-learning, human-robot cooperation, quality assurance. Dataset annotations follows hierarchical structure: task->primitive task->atomic action. Annotations further include bounding boxes for hand, parts, tools. Dataset also covers progressive proficiency in assembly tasks. Baselines are evaluated on tasks such as object recognition, action recognition, etc.

* Also, to clarify, although the paper claims this is the first work to introduce assembly video understanding, it is actually not, there are various prior work which address this (as also discussed in the paper). Paper uses a lot of superlatives, when in many cases, they are actually not true. Paper is good as it is, going overboard with overclaims and superlatives might not be a good idea in my opinion. It gives a feel of reading cheap blog post rather than a good research paper, striking a balance might be suggested.

---

> ### Author Response · Authors · 2023-08-21
> **Response to Reviewer HHfr**
>
> Thank you for your feedback on our work. We appreciate the recognition of the usefulness of our work.
>
> > *"overclaims and use of superlatives should be adjusted as mentioned in summary."*
>
> Thank you for your recognition of our work. We understand your concerns about overclaims and the use of superlatives in our previous manuscript.  The intended contribution we wanted to claim was that HA-ViD features: (1) diverse collaboration patterns of real-world assembly, (2) natural human behavior and learning progression during assembly, and (3) systematically annotate actions using subject, action verb, manipulated object, target object, and tool. Aligned with our contributions, Table 1 shows a comparison of HA-ViD with other datasets on these features. To address your concerns, we have revised the Introduction and Abstract to clarify the contribution of our work and clear up any overclaims.
>
> > *"L199 this insight is not clear. Can authors please elaborate?"*
>
> Thank you for bringing this point to our attention. We apologize for any ambiguity in Insight 4 of the previous manuscript. In Insight #4, we discuss the applications enabled by our wrong action annotations. We believe our wrong action annotations enable three key research directions.
>
> 1. Gain insight into human learning and performance.
> 2. Identify wrong actions (wrong position or order) for quality assurance.
> 3. Understand how humans fix errors and re-plan the assembly sequence.
>
> We have revised the Manuscript to enhance the clarity of Insight #4. The changes can be found in Line 208.

---

### Author Response · Authors · 2023-06-18
**HA-ViD Data Access**

To ensure the security and confidentiality of the human participant data within HA-VID, access to the dataset is protected. In order to gain access to the dataset, interested parties are kindly asked to read and agree with the Code of Conduct and request access on our website (link: https://iai-hrc.github.io/ha-vid#request_access).

For the reviewers of our paper, we have provide exclusive access to the HA-VID dataset. Please find the necessary details below:

Dataset Access Link:
https://www.dropbox.com/sh/zmm0j4xih5vq9jk/AAAqlpcMKbJuPrqxLCs4wjiRa?dl=0

Password:
havid760

Should you require any further assistance or encounter any difficulties accessing the dataset, please do not hesitate to contact us.

Thank you for your cooperation and understanding.

Best regards,
Authors of HA-ViD

---

### Author Response · Authors · 2023-08-21
**Global Response**

We would like to thank the Program Chairs, Area Chairs, and Reviewers for their diligent work. We are genuinely committed to enhancing the quality of our work and appreciate the reviewers’ insightful comments and suggestions. We have individually responded to each reviewer and have made the necessary changes to the Manuscript and Supplementary. The changes are marked in red in the updated Manuscript and Supplementary we have uploaded.

Our revisions are summarized below:
1. To enhance the robustness of our benchmarks, we have benchmarked three more recent models for action recognition and action segmentation.
2. Taking a step to investigate more advanced assembly video reasoning tasks on our dataset, we have added a benchmark for the action anticipation task.
3. To provide more practical insights, we have employed an oversampling strategy to address the class imbalance issue of our dataset.
4. We have enriched our insights (Section 3.2 to 3.5) on integrating video understanding and reasoning tasks for comprehending application-oriented assembly knowledge.
5. We have added a Discussion section (Section 4) to discuss the utilization, limitation, and future work of our dataset toward real-world applications.
6.	We have added a new table in the Supplementary comparing the quantitative statistics of our dataset with existing assembly video datasets.
7.	We have made textual and visual refinements to our Manuscript and Supplementary to improve clarity.

The authors are happy to respond to any further comments the Program Chairs, Area Chairs and Reviewers may raise hereafter.

---

### Decision · Program_Chairs · 2023-09-22

**Decision:**

Accept (Poster)

**Comment:**

The submission received unanimous support from all reviewers. The AC agrees. The authors are encouraged to revise the submission based on the reviews in the camera-ready version.